# A Statistical Framework for Analyzing Specification Resistance to Learnware-Inversion Risks

**Hao-Yi Lei** [1 2]    **Zhi-Hao Tan** [1 2]    **Zhi-Hua Zhou** [1 2]

## Abstract

The learnware paradigm enables model reuse by pairing each submitted model with a specification, a public artifact used to identify helpful models without raw-data exchange. This design creates a privacy surface: a useful specification must reveal capability-relevant information, but such information should not expose sensitive properties of training data or user tasks. Is it achievable in practice? To answer this question, this paper establishes the first framework for analyzing the incremental risk introduced by specifications in learnware, and provides theoretical guarantees for the widely used reduced kernel mean embedding (RKME) specification. Specifically, we formulate learnware-inversion as a family of statistical decision games and define the risk of specification as the incremental Bayes value from observing the model alone to observing the complete learnware. For the RKME specification, we derive risk bounds through an RKHS-smoothed total-variation bridge and the stability analysis of its reduced-set generator. We further instantiate the framework for common attacks and show that a properly sized RKME specification introduces negligible additional privacy risk while retaining sufficient information for learnware identification.

## 1. Introduction

The increasing availability of trained models makes model reuse an appealing alternative to training a new model from scratch. In many scenarios, developers may already possess models that encode useful knowledge, while future users may have only limited data, computation, or expertise. The main obstacle is not merely whether these models can be collected, but whether a model repository can identify which models are useful for a particular new task. Coarse descriptions such as model names, architectures, tags, or benchmark scores are usually insufficient to characterize reusable capability. Direct evaluation on the user's raw data is more informative, but may be expensive and privacy-sensitive. Revealing the developer's training data would also help model identification, but is rarely acceptable due to privacy and proprietary concerns. Thus, effective model reuse requires a public signal of capability that is informative enough for search, but does not rely on raw-data exchange.

The learnware paradigm provides such a mechanism through the principle of "Learnware = Model + Specification" (Zhou, 2016; Zhou & Tan, 2024): each reusable model is paired with a specification that summarizes its capability for future tasks. During submission, a developer uploads the model and its specification to a learnware dock system. During deployment, a user expresses her task requirement, also through a specification, and the system searches for helpful learnwares by comparing specifications. In this way, the specification serves as the public artifact through which learnwares become identifiable and reusable. Among the existing specifications, RKME specification has become a central choice. It sketches the developer's task distribution by a compact weighted reduced set and has supported a series of studies on practical dock systems, heterogeneous feature spaces, efficient identification, and learnware identification (Tan et al., 2024b;a; Liu et al., 2024; 2025).

The usefulness of a specification also makes its privacy analysis necessary. A specification is released publicly and is typically generated from data, distribution, model behavior, or task information. Hence, while it should preserve enough capability-relevant information for learnware identification, it should also not expose sensitive properties of the developer's training data or the user's task. Recently, Lei et al. (2024) studied the privacy-preserving ability of RKME specifications from the viewpoint of releasing the specification itself, and showed that RKME can resist several common attacks in that setting. In a learnware system, however, the released object is the full learnware: the adversary observes both the model and its specification. Since the model may already reveal some information about its training data, the relevant question is not only what the specification reveals

[1] National Key Laboratory for Novel Software Technology, Nanjing University, Nanjing, Jiangsu, China [2] School of Artificial Intelligence, Nanjing University, Nanjing, Jiangsu, China . Correspondence to: Zhi-Hua Zhou <zhouzh@nju.edu.cn>.

*Proceedings of the 43rd International Conference on Machine Learning*, Seoul, South Korea. PMLR 306, 2026. Copyright 2026 by the author(s).

in isolation, but how much additional inference power it contributes beyond the model alone. This incremental viewpoint is the central privacy question studied in this paper.

This question calls for a formal theory of specification risk in full learnware. Existing privacy analyses mainly focus on randomized data-release mechanisms such as differential privacy (Dwork, 2006), or on model-side leakage through membership inference and model inversion attacks (Shokri et al., 2017; Yeom et al., 2018; Fredrikson et al., 2014; 2015). In contrast, full learnware creates an incremental privacy interface: the model is already public, and the specification is attached as an additional utility-preserving artifact. A theory for this setting should separate model-side information from the specification-side contribution, and should explain how this contribution can be controlled for a concrete specification mechanism. In this paper, we take a first step in this direction for the widely used RKME specification. The main contributions of this work are summarized as follows:

- We formulate learnware-inversion as a family of statistical decision games and define the *risk of specification* as the incremental Bayes value from the model-only observation to the full-learnware observation. This formulation treats the released model as the baseline and the specification as additional information, yielding exact Bayes characterizations and a conditional-influence certificate for analyzing specification-side risk.

- We connect the conditional-influence certificate to RKME specifications. Through an RKHS-smoothed total-variation bridge, we reduce specification-induced distinguishability to perturbations of the RKME mean element, and further control these perturbations by analyzing RKME as a reduced-set projection in RKHS.

- We instantiate the framework for several representative learnware-inversion attacks including membership inference and finite attribute inference. The resulting bounds show that, under natural RKME deployment conditions where the reduced-set solution is locally well-conditioned, the additional risk can be reduced to a negligible term by choosing an appropriate RKME size while retaining sufficient information for learnware identification. This further yields a principled trade-off between identification accuracy and specification privacy risk.

## 2. Preliminaries and Risk of Specification

In this section, we introduce the learnware setting and the RKME specification used in the paper. We then formalize learnware-inversion and define the risk of specification.

### 2.1. Learnware and RKME Specifications

Let $\mathcal{X}$ be the feature space, $\mathcal{Y}$ be the label space, and $\mathcal{Z} = \mathcal{X} \times \mathcal{Y}$ be the sample space. A developer holds a training dataset $D = (S_1, \ldots, S_n) \in \mathcal{Z}^n$, where $S_i = (X_i, Y_i)$ are sampled from an underlying distribution $P$ over $\mathcal{Z}$. A training algorithm $\mathcal{A}_{\mathrm{tr}}$ produces a model $f = \mathcal{A}_{\mathrm{tr}}(D)$, where possible randomness of the training algorithm is included in the probability space. In the learnware paradigm, the submitted learnware is a pair $L_f = (f, R_f)$, where $R_f$ is a specification summarizing the reusable capability of $f$.

**Submitting stage.** In this stage, model developers submit their learnwares to the learnware dock system. For a developer-side model $f$, the specification $R_f$ is generated from the dataset $D$ associated with the model and is used by the system as the public capability representation of $f$.

**Deploying stage.** In this stage, a user has a new learning task and a task dataset $D_u$. The user submits a requirement specification $R_u$ generated from $D_u$, and the system identifies helpful learnwares by comparing $R_u$ with the stored specifications $\{R_f\}$ of the models in the learnware dock system. The returned learnwares can then be reused to solve the user's task. Thus, the specification is the public interface for learnware identification and reuse.

This paper focuses on the reduced kernel mean embedding (RKME) specification. Let $k : \mathcal{X} \times \mathcal{X} \to \mathbb{R}$ be a positive definite kernel, let $\mathcal{H}$ be its reproducing kernel Hilbert space (RKHS), and write $\varphi(x) = k(x, \cdot) \in \mathcal{H}$ for the canonical feature map. Since RKME is generated from the feature samples, we denote the feature part of $D$ by $D_X = (X_1, \ldots, X_n)$ and write its empirical kernel mean embedding as $\widehat{\mu}_D = n^{-1} \sum_{i=1}^n \varphi(X_i) = n^{-1} \sum_{i=1}^n k(X_i, \cdot)$.

For a prescribed specification size $m$, an RKME specification is represented by a weighted reduced set $R_f = \{(\beta_j, z_j)\}_{j=1}^m$, where $\beta_j \in \mathbb{R}$ and $z_j \in \mathcal{X}$. It induces the reduced mean element $\widetilde{\mu}_{R_f} = \sum_{j=1}^m \beta_j k(z_j, \cdot)$. Equivalently, for $\theta = (\beta_1, \ldots, \beta_m, z_1, \ldots, z_m)$, define $\Phi_m(\theta) = \sum_{j=1}^m \beta_j k(z_j, \cdot)$. The RKME generator returns a reduced set by solving the reduced-set approximation problem

$$J_D(\theta) = \frac{1}{2} \left\| \widehat{\mu}_D - \Phi_m(\theta) \right\|_{\mathcal{H}}^2. \tag{1}$$

The returned reduced set is used as the released specification $R_f$, which captures the distribution information. Eq. (1) is nonlinear and non-convex, so our analysis treats the RKME specification as the output of the reduced-set generation procedure and studies its stability at regular local optima. More details on RKME are provided in Appendix B.1.

The utility of RKME for learnware identification is determined by how well $\widetilde{\mu}_{R_f}$ approximates the task distribution in RKHS. This is mainly because a learnware and a user requirement can be compared through distances between their RKHS distances of RKME. The specification is generated from the developer's data and released together with the model, which motivates the privacy formulation below.

## 2.2. Methodology for Formalizing Specification Risk

A learnware-inversion attack is an inference procedure that uses the released learnware interface to predict a hidden target related to the developer's data. The target may encode membership, a sensitive attribute, or another privacy-relevant property. We formalize such attacks at the level of privacy games, so that different adversarial goals and side information can be handled in a common notation.

At an operational level, a learnware-inversion attack on $(f, R_f)$ can be described by an adversary $A$, a hidden target $T$, and side information $\nu$. The challenger generates the learnware and the target, reveals the corresponding observation to the adversary, and the adversary outputs a guess of $T$. The risk of the specification is then measured by comparing the optimal adversary observing the full learnware with the optimal adversary observing the model-only interface.

Throughout the paper, all sample, specification, and observation spaces are assumed to be standard Borel spaces, so that regular conditional distributions are well defined.

**Definition 2.1** (Learnware-inversion game). Fix a data distribution $P$, a training size $n$, a training algorithm $\mathcal{A}_{\mathrm{tr}}$, and a specification generator $\mathcal{R}_m$. A finite-target learnware-inversion game is a pair $\mathcal{G} = (\mathcal{T}, \mathsf{C})$, where $\mathcal{T}$ is a finite target space and $\mathsf{C}$ is a challenge kernel that generates a target-side-information pair $(T, \nu)$ with $T \in \mathcal{T}$. The interaction proceeds as follows:

(1) The challenger samples $D \sim P^n$ and generates $f = \mathcal{A}_{\mathrm{tr}}(D)$ and $R_f = \mathcal{R}_m(D)$.

(2) The challenger samples $(T, \nu) \sim \mathsf{C}(D, P)$ and reveals either the full-learnware observation $O_1 = (f, R_f, \nu)$ or the model-only observation $O_0 = (f, \nu)$.

(3) The adversary observes $O_j$, $j \in \{0, 1\}$, and outputs a prediction $\widehat{T}_j \in \mathcal{T}$.

The challenge kernel $\mathsf{C}$ specifies the privacy target and the side information. For example, it may generate a membership bit, a sensitive attribute, or a finite latent property of a record. This formulation keeps the learnware-inversion interface general, while concrete privacy targets are obtained by choosing different challenge kernels.

An adversary observing $O_j$ is a randomized decision rule $A_j$ measurable with respect to $O_j$. If $\widehat{T}_j \sim A_j(\cdot \mid O_j)$, its gain in game $\mathcal{G}$ is

$$\mathrm{gain}_{\mathcal{G}}(A_j; O_j) = \Pr_{\mathcal{G}}\left[\widehat{T}_j = T\right], \qquad (2)$$

where the probability is over the randomness of the dataset, training algorithm, specification generator, challenge kernel, and adversary. The model-only observation $O_0$ is the baseline, and the full-learnware observation $O_1$ is the refinement obtained by adding the specification.

**Definition 2.2** (Risk of specification). For a learnware-inversion game $\mathcal{G}$, define the optimal model-only gain and full-learnware gain by $G_0^\star(\mathcal{G}) = \sup_{A_0} \mathrm{gain}_{\mathcal{G}}(A_0; O_0)$ and $G_1^\star(\mathcal{G}) = \sup_{A_1} \mathrm{gain}_{\mathcal{G}}(A_1; O_1)$, where the suprema range over all randomized decision rules measurable with respect to the corresponding observations. The *risk of specification* in game $\mathcal{G}$ is

$$\mathrm{Risk}_{\mathcal{G}}(R_f \mid f) = G_1^\star(\mathcal{G}) - G_0^\star(\mathcal{G}). \qquad (3)$$

This definition isolates the incremental privacy risk caused by the specification. The released model and side information form the baseline view $O_0$, while $R_f$ provides the additional public artifact in $O_1$. Since $O_0$ is a measurable function of $O_1$, a full-learnware adversary can ignore $R_f$ and simulate any model-only adversary; hence $\mathrm{Risk}_{\mathcal{G}}(R_f \mid f) \geq 0$.

The definition is target-dependent, as privacy risk is meaningful only after specifying what is to be inferred. For a prescribed family $\mathfrak{G}$ of learnware-inversion games, we use the target-family risk $\mathrm{Risk}_{\mathfrak{G}}(R_f \mid f) = \sup_{\mathcal{G} \in \mathfrak{G}} \mathrm{Risk}_{\mathcal{G}}(R_f \mid f)$. In this paper, the main families considered later are membership inference games, finite attribute inference games, and posterior-style learnware-inversion games.

We next record two canonical challenge kernels that instantiate the general definition. For RKME-based learnware, these attacks are particularly natural. Since RKME releases reduced points in the same feature space and preserves distributional information for search, the most direct privacy concerns are record participation and sensitive-attribute exposure. We therefore record membership inference and finite attribute inference as two canonical challenge kernels; posterior-style attacks can be obtained later by changing the challenge kernel and the adversary's decision rule.

**Membership inference.** The membership challenge asks whether a candidate record comes from the training dataset or from the population. After $D = (S_1, \ldots, S_n)$ is sampled, the challenger draws $B \sim \mathrm{Bernoulli}(1/2)$. If $B = 1$, it samples $I \sim \mathrm{Unif}([n])$ and sets $S^\star = S_I$; if $B = 0$, it samples an independent fresh record $S^\star \sim P$. The target is $T = B$ and the side information is $\nu = S^\star$. The adversary observes either $(f, S^\star)$ or $(f, R_f, S^\star)$ and predicts whether the challenge record came from the training set. This source-bit formulation is well defined for both discrete and continuous data distributions.

**Finite attribute inference.** The attribute challenge asks the adversary to recover a sensitive component of a record from non-sensitive information. Let $a : \mathcal{Z} \to \mathcal{A}$ be a finite-valued sensitive attribute and let $b : \mathcal{Z} \to \mathcal{B}$ denote the non-sensitive information revealed to the adversary. For example, if $S = (X, Y)$ and $X = (X^i, X^{-i})$, one may

take $a(S) = X^i$ and $b(S) = (X^{-i}, Y)$. In the record-level attribute game, the challenger samples $I \sim \mathrm{Unif}([n])$, sets $S^\star = S_I$, and defines $T = a(S^\star)$ and $\nu = b(S^\star)$. The adversary observes either $(f, \nu)$ or $(f, R_f, \nu)$ and predicts the sensitive attribute $T$.

## 3. Decision-Theoretic Specification Risk

Section 2 defines the risk of specification through the optimal gain of learnware-inversion games. We now turn this operational definition into a Bayes decision value. This gives an exact characterization of the optimal adversary and identifies the conditional influence quantity that will later be controlled for RKME specifications. All proofs in this and the following sections are deferred to Appendix D–F.

Throughout this section, we fix a finite-target learnware-inversion game $\mathcal{G} = (\mathcal{T}, \mathsf{C})$. For brevity, we denote $\Delta_\mathcal{G} = \mathrm{Risk}_\mathcal{G}(R_f \mid f)$. Recall that the model-only and full-learnware observations are $O_0 = (f, \nu)$ and $O_1 = (f, R_f, \nu)$. Let $\mathcal{F}_j = \sigma(O_j)$ for $j \in \{0, 1\}$. Since $O_0$ is determined by $O_1$, we have $\mathcal{F}_0 \subseteq \mathcal{F}_1$. Thus releasing the specification is an information refinement.

### 3.1. Bayes Value and Functional Influence

We first characterize the optimal gain in a learnware-inversion game. For any sub-$\sigma$-field $\mathcal{F}$, define the posterior vector $\eta_\mathcal{F}(t) = \Pr(T = t \mid \mathcal{F})$ for $t \in \mathcal{T}$, and let $\Phi(p) = \max_{t \in \mathcal{T}} p_t$ for $p$ in the probability simplex over $\mathcal{T}$. The Bayes value of observing $\mathcal{F}$ is

$$\mathcal{V}_\mathcal{G}(\mathcal{F}) = \mathbb{E}\left[\max_{t \in \mathcal{T}} \Pr(T = t \mid \mathcal{F})\right]. \tag{4}$$

For an observation $O$, we also write $\mathcal{V}_\mathcal{G}(O) = \mathcal{V}_\mathcal{G}(\sigma(O))$.

The following theorem shows that the supremum over adversaries in Definition 2.2 is exactly the Bayes value.

**Theorem 3.1** (Bayes value of learnware-inversion). *For any learnware-inversion game $\mathcal{G}$, the optimal gains satisfy*

$$G_j^\star(\mathcal{G}) = \mathcal{V}_\mathcal{G}(O_j), \qquad j \in \{0, 1\}.$$

*Consequently,*

$$\Delta_\mathcal{G} = \mathcal{V}_\mathcal{G}(O_1) - \mathcal{V}_\mathcal{G}(O_0). \tag{5}$$

*Moreover, if $T \in \{0, 1\}$ and $\Pr(T = 0) = \Pr(T = 1) = 1/2$, then for any observation $O$,*

$$\mathcal{V}_\mathcal{G}(O) = \frac{1}{2}\left[1 + \mathrm{TV}\left(P_{O|1}, P_{O|0}\right)\right], \tag{6}$$

*where $P_{O|t} = \mathcal{L}(O \mid T = t)$.*

The theorem removes the explicit optimization over all adversaries. The optimal learnware-inversion adversary is

the Bayes rule induced by its observation. In the balanced binary case, its success probability is determined by the total-variation distinguishability between two observations.

Since $\mathcal{F}_0 \subseteq \mathcal{F}_1$, we have $\eta_{\mathcal{F}_0} = \mathbb{E}[\eta_{\mathcal{F}_1} \mid \mathcal{F}_0]$. Because $\Phi$ is convex, Eq. (5) also gives $\Delta_\mathcal{G} \geq 0$ by Jensen's inequality. Thus specification risk is a value-of-information gap: the full-learnware view can refine the model-only posterior, but never decreases the optimal Bayes value.

This also gives a statistical-experiment view of specification risk. For each privacy game $\mathcal{G}$, the model-only observation $O_0$ and the full-learnware observation $O_1 = (O_0, R_f)$ define two experiments for inferring the same target $T$. Since $O_1$ refines $O_0$, it is more informative in the sense of Blackwell's value ordering: for every game $\mathcal{G}$, $\mathcal{V}_\mathcal{G}(O_1) \geq \mathcal{V}_\mathcal{G}(O_0)$. For a prescribed family $\mathfrak{G}$ of privacy games, the aggregate specification value can therefore be written as

$$\Delta_\mathfrak{G}(O_1, O_0) = \sup_{\mathcal{G} \in \mathfrak{G}} \left\{\mathcal{V}_\mathcal{G}(O_1) - \mathcal{V}_\mathcal{G}(O_0)\right\}. \tag{7}$$

Thus, specification risk can be interpreted as a restricted one-sided value-of-information gap between two statistical experiments: the bare-model experiment and the full-learnware experiment. We next examine the local structure of this refinement. Let $\eta_j = \eta_{\mathcal{F}_j}$ and $H = \eta_1 - \eta_0$. Then $\mathbb{E}[H \mid \mathcal{F}_0] = 0$, and $\Delta_\mathcal{G} = \mathbb{E}\left[\Phi(\eta_0 + H) - \Phi(\eta_0)\right]$.

The next theorem identifies the infinitesimal influence of this posterior perturbation.

**Theorem 3.2** (Functional influence and decision boundary). *The map $\Phi(p) = \max_{t \in \mathcal{T}} p_t$ is Hadamard directionally differentiable. For any direction $h$ with $\sum_t h_t = 0$,*

$$\Phi_p'(h) = \max_{t \in M(p)} h_t, \qquad M(p) = \arg\max_{t \in \mathcal{T}} p_t.$$

*For the specification perturbation $H = \eta_1 - \eta_0$,*

$$\left.\frac{d}{d\epsilon}\right|_{\epsilon=0+} \mathbb{E}[\Phi(\eta_0 + \epsilon H)] = \mathbb{E}\left[\max_{t \in M(\eta_0)} H_t\right].$$

*In particular, if the model-only posterior has a unique maximizer almost surely, then this first-order derivative is zero.*

*For binary $T$, let $\pi_j = \Pr(T = 1 \mid \mathcal{F}_j)$, $a = \pi_0 - \frac{1}{2}$, and $\delta = \pi_1 - \pi_0$. Then*

$$\Delta_\mathcal{G} = \mathbb{E}\left[|a + \delta| - |a|\right]. \tag{8}$$

*Conditionally on $\mathcal{F}_0$, the contribution on $\{a \neq 0\}$ is*

$$2\,\mathbb{E}\left[\left(-\mathrm{sgn}(a)(a + \delta)\right)_+ \mid \mathcal{F}_0\right],$$

*and the contribution on the boundary $\{a = 0\}$ is $\mathbb{E}[|\delta| \mid \mathcal{F}_0]$.*

The theorem gives a geometric view of specification risk. Away from ties in the model-only posterior, the first-order

effect of the specification vanishes because the posterior perturbation has conditional mean zero. A positive gain is created when the specification moves the posterior across the model-only Bayes decision boundary, or when the model-only view is already near such a boundary. This local result motivates a conditional, model-aware measure of the specification's marginal contribution.

### 3.2. Conditional Influence Certificates

The functional influence above describes the local behavior of Bayes value under an infinitesimal posterior perturbation. For RKME analysis, we need a non-asymptotic certificate that upper bounds the finite gain in Eq. (5). For each value of $O_0$, let $\pi_t(O_0) = \Pr(T = t \mid O_0)$ and choose a measurable model-only Bayes action $t_0(O_0) \in \arg\max_{t \in \mathcal{T}} \pi_t(O_0)$. Let $P_t^{O_0} = \mathcal{L}(R_f \mid O_0, T = t)$ be the conditional law.

**Definition 3.3** (Conditional influence). The conditional influence $\mathrm{CInf}_{\mathcal{G}}(R_f \mid O_0)$ of the specification in game $\mathcal{G}$ is

$$\mathbb{E}_{O_0}\left[ \sum_{t \neq t_0(O_0)} \pi_t(O_0)\, \mathrm{TV}\left(P_t^{O_0}, P_{t_0(O_0)}^{O_0}\right) \right]. \quad (9)$$

This quantity has two components. The coefficient $\pi_t(O_0)$ measures the residual uncertainty left by the model-only view, while the total-variation term measures whether the specification can still distinguish target $t$ from the current model-only Bayes target after conditioning on $O_0$. We now show that this certificate controls the specification risk.

**Theorem 3.4** (Conditional influence bound). *For every finite-target learnware-inversion game $\mathcal{G}$,*

$$0 \leq \Delta_{\mathcal{G}} \leq \mathrm{CInf}_{\mathcal{G}}(R_f \mid O_0). \quad (10)$$

*In particular, if $T \in \{0, 1\}$ and $\pi(O_0) = \Pr(T = 1 \mid O_0)$, then*

$$\begin{aligned} \mathrm{CInf}_{\mathcal{G}}(R_f \mid O_0) = \mathbb{E}_{O_0}\Big[ &\min\{\pi(O_0), 1 - \pi(O_0)\} \\ &\cdot \mathrm{TV}\left(P_1^{O_0}, P_0^{O_0}\right)\Big]. \end{aligned} \quad (11)$$

This theorem is the main decision-theoretic certificate used in the rest of the paper. It separates the model and the specification in a precise way. If the model-only view already determines the target, the residual uncertainty term is small. If the specification behaves similarly under different target values after conditioning on the model-only view, the total-variation term is small. Either case limits the incremental contribution of the specification. The same idea also admits an information-theoretic form. This is useful when the specification is analyzed through conditional information rather than pairwise distinguishability.

**Corollary 3.5** (Conditional information bound). *For every finite-target learnware-inversion game $\mathcal{G}$,*

$$\Delta_{\mathcal{G}} \leq \mathbb{E}\left[ \mathrm{TV}\left(\mathcal{L}(T \mid O_1), \mathcal{L}(T \mid O_0)\right) \right]. \quad (12)$$

*Consequently,*

$$\Delta_{\mathcal{G}} \leq \sqrt{\frac{1}{2} I(T; R_f \mid O_0)}. \quad (13)$$

The corollary shows that the specification contributes little to any finite-target learnware-inversion game when it carries little conditional information about the target after the released model and side information are fixed. This bound is weaker than the conditional influence certificate in attack-specific analysis, but it provides a compact information-theoretic interpretation of specification risk.

The results in this section are mechanism-agnostic. They apply to any specification generator and to any released model interface included in $O_0$. Their role is to identify what must be controlled: the conditional distinguishability induced by $R_f$ after the model-only view is fixed.

## 4. RKME Influence via Projection Stability

Section 3 shows that the incremental risk of a specification is governed by the conditional distinguishability of $R_f$ after the model-only view has been fixed. We now specialize this certificate to RKME specifications. The key point is that RKME is not an arbitrary released object: it is generated by projecting the empirical kernel mean embedding onto a finite reduced-set family in RKHS. The stability of this projection controls how much one training record can perturb the released specification, and hence controls the RKHS-resolution distinguishability induced by RKME.

Throughout this section, write $\widetilde{\mu}_D = \widetilde{\mu}_{\mathcal{R}_m(D)}$ for the reduced mean element released by the RKME generator on dataset $D$. We assume that $\mathcal{X} \subseteq \mathbb{R}^d$, the kernel $k$ is twice continuously differentiable in a neighborhood of the returned reduced points, and $k(x, x) \leq \kappa^2$ for all $x \in \mathcal{X}$.

### 4.1. Projection Geometry and RKME Stability

Recall from Eq. (1) that RKME solves a reduced-set approximation problem in RKHS. Let $\Theta_m$ be the parameter space of reduced sets, with $\theta = (\beta_1, \ldots, \beta_m, z_1, \ldots, z_m)$, and define $\Phi_m(\theta) = \sum_{j=1}^m \beta_j k(z_j, \cdot)$. The family of all reduced RKME mean elements is

$$\mathcal{M}_m = \{\Phi_m(\theta) : \theta \in \Theta_m\} \subseteq \mathcal{H}.$$

Thus RKME can be viewed locally as a metric projection of $\widehat{\mu}_D$ onto the reduced-set family $\mathcal{M}_m$.

For any $\mu \in \mathcal{H}$, define the population version of the RKME objective by $J_\mu(\theta) = \frac{1}{2}\|\mu - \Phi_m(\theta)\|_{\mathcal{H}}^2$. For a dataset $D$, we have $J_D = J_{\widehat{\mu}_D}$. Let $\theta_D$ be the reduced-set parameter returned by the generator, and set $r_D = \widehat{\mu}_D - \Phi_m(\theta_D)$.

We first record the curvature certificate that turns the RKME optimization problem into a stable projection problem.

**Definition 4.1** (Regular RKME solution). A RKME solution $\theta_D$ is called regular if it is a stationary point of $J_{\widehat{\mu}_D}$ and its curvature-residual margin is positive. More precisely, let

$$s_D^2 = \inf_{\|u\|=1} \|D\Phi_m(\theta_D)[u]\|_{\mathcal{H}}^2, \quad \rho_D = \|r_D\|_{\mathcal{H}}.$$

Let $M_D = \sup_{\|u\|=1} \|D^2\Phi_m(\theta_D)[u,u]\|_{\mathcal{H}}$. The margin is

$$\chi_D = s_D^2 - \rho_D M_D. \tag{14}$$

The solution is regular if $\chi_D > 0$.

The term $s_D$ measures the nondegeneracy of the tangent directions of the reduced-set family, while $\rho_D M_D$ measures how much the residual interacts with the curvature of $\mathcal{M}_m$. Thus $\chi_D > 0$ says that the local quadratic curvature of the RKME objective dominates the curvature-residual correction. We next make this statement explicit.

**Theorem 4.2** (Curvature certificate for RKME). *Let $\theta_D$ be a stationary point of $J_{\widehat{\mu}_D}$. The Hessian of $J_{\widehat{\mu}_D}$ at $\theta_D$ satisfies, for every parameter direction $u$,*

$$\nabla^2 J_{\widehat{\mu}_D}(\theta_D)[u,u] = \|D\Phi_m(\theta_D)[u]\|_{\mathcal{H}}^2 \\ - \langle r_D, D^2\Phi_m(\theta_D)[u,u]\rangle_{\mathcal{H}}. \tag{15}$$

*Consequently, if $\chi_D > 0$, then $\nabla^2 J_{\widehat{\mu}_D}(\theta_D) \succeq \chi_D I$.*

This theorem gives a verifiable analytic certificate for RKME regularity. The stability of the returned specification follows from the local curvature of the objective at the returned point. We next derive the local stability of the RKME generator. Let $\mathcal{T}_m$ denote the local reduced-set projection functional, i.e., $\mathcal{T}_m(\mu) = \Phi_m(\theta(\mu))$, where $\theta(\mu)$ is the regular local solution of $J_\mu$ near $\theta_D$.

**Theorem 4.3** (Local projection stability). *Assume that $\theta_D$ is regular. Then there exists a neighborhood of $\widehat{\mu}_D$ on which the local RKME projection functional $\mathcal{T}_m$ is continuously differentiable. Moreover,*

$$D\mathcal{T}_m(\widehat{\mu}_D)[h] = D\Phi_m(\theta_D)H_D^{-1}D\Phi_m(\theta_D)^*h, \tag{16}$$

*where $H_D = \nabla^2 J_{\widehat{\mu}_D}(\theta_D)$, and*

$$\|D\mathcal{T}_m(\widehat{\mu}_D)\|_{\mathrm{op}} \le C_{\mathrm{stab}}(D,m) := \frac{\|D\Phi_m(\theta_D)\|_{\mathrm{op}}^2}{\chi_D}.$$

*Thus, for any two empirical mean embeddings $\mu, \mu'$ in this neighborhood, denote $\Delta\mathcal{T} := \|\mathcal{T}_m(\mu) - \mathcal{T}_m(\mu')\|_{\mathcal{H}}$,*

$$\Delta\mathcal{T} \le C_{\mathrm{stab}}(D,m)\|\mu - \mu'\|_{\mathcal{H}} + o(\|\mu - \mu'\|_{\mathcal{H}}). \tag{17}$$

*In particular, if $D$ and $D'$ differ in one record and $\widehat{\mu}_{D'}$ remains in the same regularity neighborhood, then*

$$\|\widetilde{\mu}_D - \widetilde{\mu}_{D'}\|_{\mathcal{H}} \le \frac{2\kappa C_{\mathrm{stab}}(D,m)}{n} + o(n^{-1}). \tag{18}$$

The theorem explains the record-level stability of RKME. Replacing one sample changes the empirical KME by at most $2\kappa/n$ in RKHS norm; the RKME generator amplifies this perturbation by the local stability factor $C_{\mathrm{stab}}(D,m)$. Geometrically, the margin $\chi_D$ plays the role of a local reach or conditioning parameter of the reduced-set family $\mathcal{M}_m$.

The same projection viewpoint yields a statistical influence function for RKME. This gives a first-order expansion of how individual samples contribute to the specification.

**Theorem 4.4** (RKME influence function). *Let $\mu_P = \mathbb{E}_{X\sim P_X}k(X,\cdot)$ be the population KME of the feature marginal, and suppose that the population reduced-set projection $\widetilde{\mu}_P = \mathcal{T}_m(\mu_P)$ is regular. Then $\mathcal{T}_m$ is Hadamard differentiable at $\mu_P$, and*

$$\widetilde{\mu}_D - \widetilde{\mu}_P = \frac{1}{n}\sum_{i=1}^n \mathrm{IF}_m(X_i; P) + o_p(n^{-1/2}),$$

*where*

$$\mathrm{IF}_m(x; P) = D\mathcal{T}_m(\mu_P)[k(x,\cdot) - \mu_P]. \tag{19}$$

*Equivalently, if $H_P = \nabla^2 J_{\mu_P}(\theta_P)$ at the population regular solution $\theta_P$, then*

$$\mathrm{IF}_m(x; P) = D\Phi_m(\theta_P)H_P^{-1}D\Phi_m(\theta_P)^*[k(x,\cdot) - \mu_P].$$

*Moreover, $\|\mathrm{IF}_m(x; P)\|_{\mathcal{H}} \le 2\kappa C_{\mathrm{stab}}(P,m)$.*

The influence function gives the record-level interpretation of RKME stability. Each sample affects the released RKME through a $1/n$-scaled perturbation, and the magnitude of this perturbation is controlled by the same curvature-dependent stability factor.

### 4.2. Smoothed Distinguishability and Risk Control

We now connect the RKME stability results to the conditional influence certificate in Section 3.2. Let $\mathfrak{Q}$ and $\mathfrak{Q}'$ be two laws over RKHS mean elements. For $\Lambda > 0$, define the RKHS-smoothed total variation by

$$\mathrm{TV}_{\mathcal{H},\Lambda}(\mathfrak{Q}, \mathfrak{Q}') = \frac{1}{2}\sup_{g\in\mathcal{G}_\Lambda} |\mathbb{E}_{\mathfrak{Q}}g(U) - \mathbb{E}_{\mathfrak{Q}'}g(U)|, \tag{20}$$

where $\mathcal{G}_\Lambda$ consists of functions $g(U) = \psi(\langle h, U\rangle_{\mathcal{H}})$ with $\|h\|_{\mathcal{H}} \le \Lambda$ and $\psi : \mathbb{R} \to [-1,1]$ being 1-Lipschitz. This class captures bounded decisions based on RKHS statistics of the released mean element. The following bridge converts RKHS perturbation into smoothed distinguishability.

**Theorem 4.5** (RKHS-smoothed total-variation bridge). *Let $\mathfrak{Q}$ and $\mathfrak{Q}'$ be laws over RKHS mean elements. Then*

$$\mathrm{TV}_{\mathcal{H},\Lambda}(\mathfrak{Q}, \mathfrak{Q}') \le \frac{\Lambda}{2}W_1^{\mathcal{H}}(\mathfrak{Q}, \mathfrak{Q}'), \tag{21}$$

where $W_1^{\mathcal{H}}$ is the 1-Wasserstein distance induced by $\| \cdot \|_{\mathcal{H}}$. In particular, for point masses at $u, v \in \mathcal{H}$,

$$\mathrm{TV}_{\mathcal{H}, \Lambda}(\delta_u, \delta_v) \leq \frac{\Lambda}{2} \|u - v\|_{\mathcal{H}}.$$

If $u = \mu_P$ and $v = \mu_Q$ are kernel mean embeddings of two distributions, this becomes

$$\mathrm{TV}_{\mathcal{H}, \Lambda} \leq \frac{\Lambda}{2} \mathrm{MMD}(P, Q).$$

This theorem is the bridge between the Bayes-risk analysis and RKME geometry. The decision-theoretic risk certificate depends on distinguishability of the specification, while RKME stability controls the RKHS distance between released mean elements. The smoothed TV connects the two.

We next define the RKHS-compatible version of the conditional influence certificate. For each $O_0$ and target value $t$, let $\mathfrak{P}_t^{O_0}$ be the conditional law of the released RKME mean element $\widetilde{\mu}_D$ given $(O_0, T = t)$. Let $t_0(O_0)$ be a model-only Bayes action as in Section 3.2. Define $\mathrm{CInf}_{\mathcal{G}}^{\mathcal{H}, \Lambda}$ as

$$\mathbb{E}_{O_0} \left[ \sum_{t \neq t_0(O_0)} \pi_t(O_0) \, \mathrm{TV}_{\mathcal{H}, \Lambda} \left( \mathfrak{P}_t^{O_0}, \mathfrak{P}_{t_0(O_0)}^{O_0} \right) \right], \quad (22)$$

where $\pi_t(O_0) = \Pr(T = t \mid O_0)$. Let $\Delta_{\mathcal{G}}^{\mathcal{H}, \Lambda}$ be the specification risk when the specification-dependent part of the adversary is restricted to the RKHS-compatible class $\mathcal{G}_\Lambda$.

**Theorem 4.6** (RKME control of conditional influence). *For any finite-target learnware-inversion game $\mathcal{G}$, the RKHS-compatible specification risk defined above satisfies*

$$\Delta_{\mathcal{G}}^{\mathcal{H}, \Lambda} \leq \mathrm{CInf}_{\mathcal{G}}^{\mathcal{H}, \Lambda}. \quad (23)$$

*Moreover, suppose that for every relevant pair of target values, the conditional datasets can be coupled by replacing one record, and the RKME solutions along the coupling are regular with stability factor at most $C_{\mathrm{stab}}(m)$. Then*

$$\Delta_{\mathcal{G}}^{\mathcal{H}, \Lambda} \leq \frac{\Lambda \kappa C_{\mathrm{stab}}(m)}{n} + o(n^{-1}). \quad (24)$$

This theorem completes the link from Section 3 to RKME. Bayes specification risk is controlled by conditional influence; under an RKHS-compatible interface, conditional influence is controlled by RKHS-smoothed distinguishability; this distinguishability is controlled by RKME projection stability. Thus the record-level contribution of an RKME specification scales up to attack-dependent constants.

The conclusion should be read together with $m$. Increasing $m$ can improve the RKME approximation of the task distribution, which benefits learnware identification. At the same time, $m$ changes the geometry of the reduced-set family through $s_D$, $M_D$, and $\rho_D$, and hence changes $C_{\mathrm{stab}}(m)$. The resulting balance between RKME approximation and projection stability will be used in later sections to explain the search–risk trade-off of specification size.

## 5. Attack Instantiations and Trade-Off

We now instantiate this route for representative learnware-inversion attacks and then discuss how the RKME size $m$ affects both search utility and incremental risk.

Throughout this section, we consider the RKHS-compatible specification interface in Section 4.2. When the adversary uses the RKME specification, the specification-dependent statistic is assumed to be generated by test functions of RKHS norm at most $\Lambda$. We also assume the regularity and coupling conditions of Theorem 4.6, and write $C_{\mathrm{stab}}(m)$ for a uniform upper bound on the local RKME stability factor over the relevant datasets.

**Loss-based membership inference.** Consider the membership game in Section 2.2, where the target is the source bit $T = B$ and the side information is the challenge sample $S^\star = (X^\star, Y^\star)$. A standard membership signal is the model loss $\ell(f(X^\star), Y^\star)$, where $\ell$ is bounded. The model-only adversary may use this loss and all information in $O_0 = (f, S^\star)$. A specification-assisted attack additionally uses an RKME statistic such as $\psi(\langle h_{S^\star}, \widetilde{\mu}_D \rangle_{\mathcal{H}})$, where $\|h_{S^\star}\|_{\mathcal{H}} \leq \Lambda$ and $\psi : \mathbb{R} \to [-1, 1]$ is 1-Lipschitz. The following result specializes the conditional-influence and stability bounds to loss-based membership inference.

**Theorem 5.1** (Loss-based membership risk). *For RKHS-compatible loss-based membership inference attacks,*

$$\Delta_{\mathrm{loss\text{-}MIA}}^{\mathcal{H}, \Lambda} \leq C_{\mathrm{loss}} \frac{\Lambda \kappa C_{\mathrm{stab}}(m)}{n} + o(n^{-1}), \quad (25)$$

*where $C_{\mathrm{loss}}$ is a constant depending only on the bounded loss and the attack decision map.*

The theorem separates model-side leakage from specification leakage. The loss of the released model may already be informative about membership, especially when the model overfits, but that information belongs to the baseline observation $O_0$. The theorem controls the additional contribution of RKME, which is small when one record has only a stable effect on the released reduced mean element.

**Posterior membership inference.** Many membership attacks use the model output to form a posterior belief about whether $S^\star$ is a training record. Let $V_{\mathrm{mem}}$ denote the model-only context, such as $S^\star$, model prediction, and loss. The model provides an energy $E_f(b; V_{\mathrm{mem}})$ for $b \in \{0, 1\}$, and the RKME specification provides a prior score $\rho_R(b \mid V_{\mathrm{mem}})$. For temperature $\gamma > 0$, define

$$q_R^\gamma(b \mid V_{\mathrm{mem}}) \propto \rho_R(b \mid V_{\mathrm{mem}}) e^{-E_f(b; V_{\mathrm{mem}})/\gamma}.$$

The model-only posterior $q_0^\gamma$ is obtained by replacing $\rho_R$ with a baseline prior $\rho_0$. This posterior membership attack is a binary instance of the following general power-posterior principle.

**Theorem 5.2** (Power-posterior stability). *Let $T \in \mathcal{T}$ be finite. For an observation context $V$, define*

$$Z_R^\gamma(V) = \sum_{t' \in \mathcal{T}} \rho_R(t' \mid V) e^{-E_f(t';V)/\gamma},$$

*where $\rho_R(\cdot \mid V)$ is a probability distribution. Let*

$$q_R^\gamma(t \mid V) = \frac{\rho_R(t \mid V) e^{-E_f(t;V)/\gamma}}{Z_R^\gamma(V)}. \quad (26)$$

*where $\rho_0(\cdot \mid V)$ is a probability distribution. Let $q_0^\gamma$ be defined analogously with $\rho_0$. If $E_f(t;V) \in [0, B_E]$ for all $t$, then*

$$\begin{aligned} \mathrm{TV}&\left(q_R^\gamma(\cdot \mid V), q_0^\gamma(\cdot \mid V)\right) \\ &\leq e^{B_E/\gamma} \, \mathrm{TV}\left(\rho_R(\cdot \mid V), \rho_0(\cdot \mid V)\right). \end{aligned} \quad (27)$$

*Consequently, if the prior shift induced by $\rho_R$ is RKHS-compatible, then*

$$\Delta_{\mathrm{post},\gamma}^{\mathcal{H},\Lambda} \leq C_{\mathrm{post}} e^{B_E/\gamma} \frac{\Lambda \kappa C_{\mathrm{stab}}(m)}{n} + o(n^{-1}), \quad (28)$$

*where $C_{\mathrm{post}}$ depends on the target space and the prior map.*

This result explains the role of the temperature. The model energy reshapes the posterior, while the specification affects it through the prior shift. The factor $e^{B_E/\gamma}$ is the stability cost of this normalization. Once the RKME-induced prior shift is controlled by RKHS-smoothed distinguishability, posterior membership attacks are controlled by the same projection-stability term.

**Finite attribute inference.** We next consider the finite attribute game in Section 2.2. Let $A = a(S^\star) \in \mathcal{A}$ be the sensitive attribute, with $|\mathcal{A}| = K$, and let $\nu = b(S^\star)$ be the non-sensitive side information. The model-only view may already reveal correlations among $\nu$, model outputs, and $A$. The risk of specification measures the additional gain after this baseline has been fixed.

A common specification-assisted attribute attack uses RKME to construct a conditional prior over $A$. Suppose this prior is obtained from nonnegative RKME-linear scores $s_R(a, \nu) = \langle h_{a,\nu}, \widetilde{\mu}_D \rangle_{\mathcal{H}}$ with $\|h_{a,\nu}\|_{\mathcal{H}} \leq \Lambda$, and assume the normalization margin $\sum_{a \in \mathcal{A}} s_R(a, \nu) \geq c_0 > 0$. Let $\rho_R(a \mid \nu) = s_R(a, \nu) / \sum_{a'} s_R(a', \nu)$.

**Theorem 5.3** (Finite attribute risk). *Under the conditions above, the sample-specific specification risk for finite attribute inference satisfies*

$$\Delta_{\mathrm{AIA}}^{\mathcal{H},\Lambda} \leq \frac{2K\Lambda\kappa C_{\mathrm{stab}}(m)}{c_0 n} + o(n^{-1}). \quad (29)$$

*If the adversary further combines the RKME prior with a bounded model energy through a power posterior of temperature $\gamma$, the right-hand side is multiplied by $e^{B_E/\gamma}$.*

Attribute inference has a population-level component. If a sensitive attribute is strongly correlated with non-sensitive information, this correlation may be useful to any adversary with sufficient background knowledge. The theorem controls the record-level incremental component caused by the released RKME specification, namely the part that changes when a single developer record changes.

**A unified view of the attacks.** The three attacks above have different targets and decision rules, but the specification-side mechanism is the same. The attack determines the target space, side information, model evidence, and prior map. The RKME contribution enters through a specification-induced prior or score, and this contribution is controlled by the chain $\Delta_{\mathcal{G}} \longrightarrow \mathrm{CInf}_{\mathcal{G}} \longrightarrow \mathrm{TV}_{\mathcal{H},\Lambda} \longrightarrow \|\widetilde{\mu}_D - \widetilde{\mu}_{D'}\|_{\mathcal{H}} \longrightarrow \frac{C_{\mathrm{stab}}(m)}{n}$. Thus membership inference, attribute inference, and posterior learnware-inversion are not separate analyses. They are concrete instances of the same decision-theoretic and RKHS-stability principle.

**RKME identification utility.** We now relate the risk to learnware identification. Let $\mu_P = \mathbb{E}_{X \sim P_X} k(X, \cdot)$ be the population KME of the feature marginal, and define the empirical RKME approximation error $\eta_m(D) = \|\widetilde{\mu}_D - \widehat{\mu}_D\|_{\mathcal{H}}$. The following standard bound separates reduced-set approximation from statistical estimation.

**Lemma 5.4** (RKME approximation error). *Assume $k(x,x) \leq \kappa^2$ for all $x \in \mathcal{X}$. Then, with probability at least $1 - \delta$ over $D \sim P^n$,*

$$\|\widetilde{\mu}_D - \mu_P\|_{\mathcal{H}} \leq \eta_m(D) + \frac{\kappa}{\sqrt{n}} + \kappa\sqrt{\frac{2\log(1/\delta)}{n}}. \quad (30)$$

Combining the attack bounds with this approximation bound gives the following summary.

**Theorem 5.5** (Search–risk decomposition). *For the RKHS-compatible learnware-inversion attacks above,*

$$\Delta_{\mathcal{G}}^{\mathcal{H},\Lambda} \leq C_{\mathcal{G}} \frac{\Lambda \kappa C_{\mathrm{stab}}(m)}{n} + o(n^{-1}), \quad (31)$$

*where $C_{\mathcal{G}}$ is an attack-dependent constant. Meanwhile, learnware identification error induced by RKME is controlled by Eq. (30). Hence $m$ influences search through $\eta_m(D)$ and influences risk through $C_{\mathrm{stab}}(m)$.*

This theorem is the main design principle. A larger $m$ can reduce $\eta_m(D)$ and preserve more distributional information for matching. At the same time, it changes the geometry of the reduced-set family $\mathcal{M}_m$. In particular, $C_{\mathrm{stab}}(m) = \frac{\|D\Phi_m\|_{\mathrm{op}}^2}{s_m^2 - \rho_m M_m}$ can increase when reduced atoms become redundant, when the tangent Gram matrix becomes ill-conditioned, or when curvature effects become strong.

**Specification size choice.** The decomposition should be read as a rule for choosing the RKME size. Increasing $m$ usually reduces the reduced-set approximation error $\eta_m(D)$ and improves learnware identification. At the same time, increasing $m$ changes the geometry of the reduced-set family and may enlarge the projection-stability factor $C_{\text{stab}}(m)$; in our bound, this affects the record-level risk through $C_{\text{stab}}(m)/n$. Thus the relevant design principle is to choose $m$ large enough for identification, but not so large that the stability cost dominates the record-level averaging effect.

The conditions behind this rule are aligned with the way RKME specifications are used in learnware. Let $\Gamma_D = D\Phi_m(\theta_D)^* D\Phi_m(\theta_D)$ be the tangent Gram matrix at the returned solution. We say that the returned Gaussian RKME solutions are *polynomially nondegenerate* up to size $m_*$ if there exist constants $c_s > 0$, $\tau \in (0,1)$, $a < 1$, and $\beta_{\max} < \infty$ such that, for all $1 \leq m \leq m_*$, $|\beta_j| \leq \beta_{\max}, \lambda_{\min}(\Gamma_D) \geq c_s m^{-a}, \rho_D M_D \leq (1-\tau)\lambda_{\min}(\Gamma_D)$.

**Lemma 5.6** (Subquadratic stability). *Consider the Gaussian kernel $k_\sigma(x,z) = \exp(-\|x - z\|^2/(2\sigma^2))$. If the returned Gaussian RKME solutions are polynomially nondegenerate up to size $m_*$, then there exists a constant $C_0 > 0$, depending only on $\sigma, d, \beta_{\max}, c_s, \tau$, such that*

$$C_{\text{stab}}(m) \leq C_0 m^{2-\xi}, \xi > 0, 1 \leq m \leq m_*. \quad (32)$$

A detailed discussion of polynomial nondegeneracy for Gaussian RKME, together with the proofs of Lemma 5.6 and Corollary 5.7, is provided in Appendix F.8.

**Corollary 5.7** (Square-root RKME size). *Let $m_* = \lfloor \sqrt{n} \rfloor$, and suppose that the returned Gaussian RKME solutions are polynomially nondegenerate up to size $m_*$. We have*

$$\Delta_{\mathcal{G}}^{\mathcal{H},\Lambda}(m_*) \leq C_{\mathcal{G}} \Lambda \kappa C_0 m_*^{-\xi} + o(n^{-1}), \xi > 0. \quad (33)$$

*Moreover, if the RKME approximation error satisfies $\eta_m(D) = O(m^{-\alpha})$ for some $\alpha > 0$, then at $m = m_*$,*

$$\|\widetilde{\mu}_D - \mu_P\|_{\mathcal{H}} = O(m_*^{-\alpha}) + O_p(m_*^{-1}). \quad (34)$$

The corollary gives the theoretical role of the square-root specification size. For Gaussian RKME specification with size of $m = \sqrt{n}$, the specification-side record-level risk is bounded by the decreasing term $O(m_*^{-\xi})$. Equivalently, for any target tolerance $\varepsilon > 0$, Eq. (33) gives $\Delta_{\mathcal{G}}^{\mathcal{H},\Lambda}(m_*) \leq \varepsilon + o(n^{-1})$ whenever $m_* \geq (C_{\mathcal{G}} \Lambda \kappa C_0/\varepsilon)^{1/\xi}$. At the same time, Eq. (34) shows that the reduced-set approximation error continues to decrease with $m_*$. In the well-conditioned case where the tangent Gram matrix is uniformly nondegenerate, i.e., $a = 0$, the record-level incremental risk is $O(m_*^{-1}) = O(n^{-1/2})$ up to attack-dependent constants. Thus the square-root size serves as a natural operating point: it enlarges the reduced set enough for learnware identification while keeping the individual-record contribution controlled by the Gaussian RKME projection geometry.

## 6. Discussions

The guarantees in this paper should be read as specification-side guarantees. Our risk definition treats the released model interface as the baseline observation $O_0$ and measures only the additional value obtained after the specification is attached. Thus, information already exposed by the model, whether through queries or parameters, is not attributed to the specification. This separation matches the learnware setting: developers intentionally release models for reuse, while the question studied here is whether the specification further increases the risk of exposing the original data.

Our analysis is tied to the natural interface through which RKME specifications are used. The RKHS-smoothed total-variation bridge captures adversaries that use the specification through kernel scores, mean-embedding distances, RKHS statistics, or posterior priors derived from reduced mean elements. The RKME stability condition is a local regularity certificate of the released reduced set, derived from the curvature and nondegeneracy of the RKME objective. Ill-conditioned reduced sets may increase $C_{\text{stab}}(m)$, which suggests using regularization or selection rules that preserve projection stability. For attribute inference, our bounds control the sample-specific incremental contribution of RKME; population-level correlations between sensitive and non-sensitive cases should be interpreted separately.

The framework can also support broader specification-side analyses. Different model interfaces simply change the baseline $O_0$, while the incremental-risk viewpoint remains unchanged. Future work may study adaptive multi-query learnware docks, architecture-specific white-box attacks, and other specification mechanisms beyond RKME.

## 7. Concluding Remarks

This paper studies the incremental privacy risk introduced by attaching an RKME specification to a released learnware model. We formalize learnware-inversion as a family of statistical decision games and define the risk of specification as the increase in Bayes value from the model-only observation to the full-learnware observation. This yields conditional influence as the central certificate for the marginal privacy contribution of the specification. For RKME, we connect this certificate to RKHS-smoothed distinguishability and analyze the reduced-set generator through projection stability in RKHS. The resulting bounds for membership inference, finite attribute inference, and power-posterior attacks show that, under RKHS-compatible interfaces, record-level incremental risk is controlled by RKME projection stability, while learnware identification utility is governed by RKME approximation error. Together, these results provide a principled theoretical foundation for designing RKME specifications that balance reuse and privacy in learnware.

## Acknowledgments

This work was supported by 111 Center (B26023) and FIDBP of MoE (JYB2025XDXM118). Tan was supported by the Postdoctoral Fellowship Program of CPSF (GZB20250396) and Jiangsu Funding Program for Excellent Postdoctoral Talent.

## Impact Statement

This paper advances the learnware field from a theoretical perspective in two ways: (i) we propose a principled theoretical framework for measuring privacy risks in learnware systems, and (ii) we provide theoretical characterizations of the privacy protection performance of specification-based interfaces. While our work may have various potential societal consequences, we do not believe that any of them requires specific emphasis here.

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

# A. Related Work

## A.1. Related Work of Learnware

The learnware paradigm was introduced to enable users to solve new tasks by reusing existing trained models rather than training new models from scratch (Zhou, 2016; Zhou & Tan, 2024). Its central design is that each submitted model is paired with a specification, which serves as a public capability representation for search, matching, and reuse. Among existing specifications, reduced kernel mean embedding (RKME) has been the most widely adopted choice. RKME represents the developer's task distribution by a compact weighted reduced set, allowing the dock system to compare a user requirement with submitted learnwares without accessing raw developer data (Wu et al., 2023). Based on this specification, a series of works have studied how to identify and reuse helpful learnwares under increasingly practical settings.

Early RKME-based learnware studies focused on matching user tasks with submitted models through distributional similarity. Zhang et al. (2021) studied learnware reuse when the user's task contains unseen parts, showing that specifications can help identify models useful beyond the user's limited data. Guo et al. (2023) considered heterogeneous label spaces, where the output spaces of developer models and user tasks may differ. For heterogeneous feature spaces, Tan et al. (2024a) studied the case where auxiliary data are available to align feature spaces, while Tan et al. (2023) further removed this auxiliary-data requirement and organized heterogeneous learnwares without accessing developers' original data. These works show that the specification is not merely metadata: it is the technical object that makes learnware identification possible across different tasks, feature spaces, and label spaces.

Efficiency and scalability have also become important in the learnware paradigm. Xie et al. (2023) proposed identifying helpful learnwares without examining the whole market, by using representative anchor learnwares to avoid exhaustive comparison. Liu et al. (2024) studied how learnware specifications and markets can evolve, so that the dock system can become more accurate and efficient as more learnwares are submitted. The Beimingwu system of Tan et al. (2024b) further implemented a practical learnware dock system and demonstrated the feasibility of large-scale specification-based model reuse. More recent work has broadened the scope of model reuse and learnware-style identification, including tabular learnwares that can be repurposed for seemingly irrelevant tasks, dual-alignment specifications (Chen et al., 2025), model ranking for pre-trained model repositories (Zhang et al., 2023), and integration of task-specific experts (Lin et al., 2026). There are also emerging specifications beyond RKME, such as PAVE-style parameter-vector specifications (Tan et al., 2025; Shi et al., 2026; Liu et al., 2026). These developments reinforce the same message: a useful model-reuse system requires a public artifact that represents model capability.

The privacy of learnware specifications has only recently been studied. Lei et al. (2024) analyzed RKME specifications from the viewpoint of releasing the specification itself, and showed that RKME can resist several common attacks in that specification-alone setting. Our work studies a different privacy surface. In a deployed learnware, the adversary observes the model together with its specification. The model may already reveal information about its training data, so the relevant question is not the total leakage of the full release, but the incremental inference power introduced by attaching the specification to the released model. This paper therefore formalizes learnware-inversion as a family of decision games, defines specification risk as an incremental Bayes value from the model-only observation to the full-learnware observation, and develops RKME-specific bounds through RKHS-smoothed distinguishability and projection stability.

## A.2. Related Work of Privacy

A large body of privacy research studies how data-dependent public releases may expose private information. Classical anonymization notions, such as $k$-anonymity (Sweeney, 2002) and $\ell$-diversity (Machanavajjhala et al., 2007), aim to protect tabular data releases by making individuals indistinguishable within groups or by enforcing diversity of sensitive attributes. Synthetic-data release is another important direction: synthetic datasets are intended to support downstream analysis while avoiding direct release of raw records (Drechsler & Reiter, 2010; Bellovin et al., 2019; Stadler et al., 2022). Generative models such as GANs and VAEs have also been used to construct synthetic records (Goodfellow et al., 2014; Kingma & Welling, 2014; Xu et al., 2019). However, later studies showed that synthetic data and generative models may still leak membership, attributes, or training examples under suitable attacks (Chen et al., 2020; Hayes et al., 2019; Stadler et al., 2022). This line of work is related to RKME because RKME also releases a compact data-dependent reduced representation. The difference is that an RKME specification is not released as a general-purpose synthetic dataset; it is a capability summary used by a learnware dock for model identification.

Differential privacy provides a standard worst-case framework for limiting the effect of one record on a randomized

release (Dwork, 2006; Dwork et al., 2006). It has been used to construct private data generators and private mean-embedding mechanisms, such as PATE-GAN and DP-MERF (Jordon et al., 2018; Harder et al., 2021; Xin et al., 2022). These methods provide strong distributional guarantees, but often require explicit noise injection. For learnware specifications, excessive perturbation may directly damage search and matching utility, since the specification must preserve capability-relevant information. Prior work has also emphasized the privacy–utility tension in differentially private learning (Alvim et al., 2018; Zhao et al., 2020). Our work takes a complementary route: rather than claiming a DP guarantee for deterministic RKME, we define the specification-side incremental risk relative to a model-only baseline and bound it under RKHS-compatible attack interfaces through the stability of the RKME reduced-set generator.

Our attack model is related to membership inference, model inversion, attribute inference, and reconstruction attacks against machine learning systems. Membership inference asks whether a sample was used in training (Shokri et al., 2017; Yeom et al., 2018; Salem et al., 2019). Model inversion and attribute inference infer sensitive attributes or representative inputs from model outputs or confidence information (Fredrikson et al., 2014; 2015). These works mainly study the information exposed by a released model or a generative system. In contrast, learnware releases contain both a model and a specification. We therefore treat the released model as the baseline observation and ask how much additional Bayes value the specification contributes. This separates model-side leakage from specification-side risk, and allows the same framework to cover membership inference, finite attribute inference, and posterior-style learnware-inversion attacks.

Dataset condensation and distillation are also relevant because they release compact data-dependent objects that may preserve training utility (Wang et al., 2018; Loo et al., 2023). These methods are usually evaluated by downstream performance and empirical privacy attacks, while our setting focuses on RKME specifications as public learnware interfaces. The central object in our analysis is not whether the reduced set reconstructs the raw dataset, but whether adding the reduced-set specification to an already released model improves an adversary's ability to infer a privacy target. This incremental viewpoint is essential for full learnware privacy, where both the model and the specification are visible.

### A.3. Related Work of Theoretical Tools

Our decision-theoretic formulation is connected to classical statistical decision theory. The Bayes value of an observation is the optimal expected utility achievable by a decision rule, and comparing observations by their decision value is closely related to Blackwell's comparison of statistical experiments (Blackwell, 1953) and Le Cam's theory of statistical experiments (Le Cam, 1986). In this paper, the model-only observation $O_0 = (f, \nu)$ and the full-learnware observation $O_1 = (f, R_f, \nu)$ define two experiments for the same privacy target. Specification risk is the restricted value-of-information gap between these two experiments, restricted to learnware-inversion privacy games. This perspective allows us to remove the explicit supremum over adversaries and express optimal attack gain as a Bayes value. For binary targets, this further yields the standard total-variation characterization of the optimal testing advantage.

The conditional influence certificate used in this paper is a non-asymptotic form of this value-of-information analysis. It measures the conditional distinguishability of the specification after the model-only view has been fixed. The local interpretation through the directional derivative of the Bayes decision functional is related to the general calculus of nonsmooth functionals and Hadamard differentiability, which underlies the functional delta method (van der Vaart & Wellner, 1996). In our setting, the decision functional is $\Phi(p) = \max_t p_t$. Its directional derivative shows that the first-order specification effect vanishes away from model-only posterior ties, while the finite conditional influence bound gives a usable certificate for later RKME analysis.

The RKME part of our analysis builds on kernel mean embeddings and integral probability metrics. Kernel mean embeddings map probability distributions into an RKHS and allow distributional comparison through inner products and RKHS norms (Smola et al., 2007; Sriperumbudur et al., 2011). The maximum mean discrepancy (MMD) is the RKHS distance between mean embeddings and is a standard tool for two-sample testing and distribution comparison (Gretton et al., 2012). RKME specifications further approximate empirical kernel mean embeddings by a compact reduced set (Burges, 1996; Scholkopf et al., 1999; Wu et al., 2023). Our RKHS-smoothed total-variation bridge can be viewed as a restricted integral probability metric: instead of allowing arbitrary measurable tests as in ordinary total variation, it only allows tests generated by bounded RKHS statistics. This matches the way RKME is used in learnware search, through kernel scores, mean-embedding distances, and priors derived from reduced mean statistics.

Finally, the projection-stability part of the analysis uses a functional view of RKME generation. The RKME generator is treated as a local projection functional from the empirical kernel mean embedding to the reduced-set family in RKHS. At a regular local optimum, the Hessian of the reduced-set objective is controlled by a curvature-residual margin, and the

implicit function theorem yields local differentiability of the projection map. This gives an RKME influence-function expansion, analogous in spirit to classical influence-function and delta-method analyses (van der Vaart & Wellner, 1996). The Wasserstein form used in the RKHS-smoothed TV bridge follows the standard transportation viewpoint on Lipschitz test functions (Villani, 2009). Together, these tools connect the statistical decision quantity in Section 3 to the geometric stability of the deterministic RKME generator in Section 4.

## B. Background

### B.1. Reduced Kernel Mean Embedding

**Kernel Mean Embedding.** KME (Smola et al., 2007) is a technique in machine learning that maps the mean of a probability distribution into a Reproducing Kernel Hilbert Space (RKHS). Given a probability distribution $\mathcal{P}$ over a domain $X$, and a kernel function $k : X \times X \to \mathbb{R}$ (Schölkopf & Smola, 2002), the KME of $\mathcal{P}$ is defined as the following:

$$\mu(\mathcal{P}) = \int k(x, \cdot) d\mathcal{P}(x)$$

where $\mu(\mathcal{P})$ represents the kernel mean embedding of the distribution $\mathcal{P}$, $x$ is an element of the domain $X$, and '·' denotes a placeholder for a second argument in the kernel function. Denote the associated RKHS as $\mathcal{H}_k$ and $\phi : x \in X \to k(x, \cdot) \in \mathcal{H}_k$ the corresponding canonical feature map. The kernel function $k$ quantifies the similarity between pairs of data points and is required to be positive definite to induce a valid RKHS.

Kernel Mean Embedding (KME) exhibits an array of beneficial properties, which contribute to its appeal as a robust method for diverse machine learning endeavors, most notably in the realm of specification. By the reproducing property, $\forall f \in \mathcal{H}_k, \langle f, \mu(\mathcal{P}) \rangle = \mathbb{E}_{\mathcal{P}}[f(X)]$, which demonstrates the notion of mean. By using characteristic kernels (Sriperumbudur et al., 2011), no information about the distribution $\mathcal{P}$ would be lost during kernel embedding, i.e. $||\mu(\mathcal{P}) - \mu(\mathcal{P}')||_{\mathbb{H}_k} = 0$ implies that $\mathcal{P} = \mathcal{P}'$. One of the most commonly used kernels is the Gaussian kernel

$$k(x, x') = exp(-\gamma ||x - x'||_2^2), \gamma > 0$$

The reproducing property of the Gaussian kernel and its characteristic as a characteristic kernel make it a widely used kernel in Kernel Mean Embedding (KME) (Gretton et al., 2012; Muandet & Schölkopf, 2013; Doran, 2013). In the learnware market, for tabular data, the RKME corresponding to the Gaussian kernel is currently in use, while for image and text data, the RKME corresponding to the Gaussian kernel can also be used after extracting embeddings (Tan et al., 2024a; 2023; Xie et al., 2023). In the main analysis, we assume a bounded and sufficiently smooth kernel so that the RKME reduced-set generator admits a local stability analysis.

In learning tasks, we often have no access to the true distribution $\mathcal{P}$, but we can use samples to estimate it. The empirical kernel mean embedding is an approximation of the KME based on a finite set of samples from a probability distribution. Given a probability distribution $\mathcal{P}$ over a domain $X$, a kernel function $k : X \times X \to \mathbb{R}$, and a set of $n$ independent and identically distributed (i.i.d.) samples $\{x_1, x_2, \ldots, x_n\}$ drawn from $P$, the empirical KME $\hat{\mu}(\mathcal{P})$ of $P$ is defined as:

$$\hat{\mu}(\mathcal{P}) = \frac{1}{n} \sum_{i=1}^{n} k(x_i, \cdot)$$

The empirical KME $\hat{\mu}$ will converge to $\mu$ in the rate of $\mathcal{O}(1/\sqrt{n})$ measured by RKHS norm $|| \cdot ||_{\mathcal{H}}$ under mild conditions (Smola et al., 2007).

**Reduced Kernel Mean Embedding.** Although the properties of KME are desirable, the computation of KME becomes challenging when there are many samples, and the calculation of KME requires access to the original data. Therefore, KME is not a specification for learnware. To address this issue, (Wu et al., 2023) introduce RKME to approximate original KME via the reduced set method, which is first used to speed up SVM prediction (Burges, 1996) and receives more comprehensive studies in (Scholkopf et al., 1999).

The idea of RKME is to find a set $(\beta_j, z_j)_{j=1}^{m}$ and compute $\sum_{j=1}^{m} \beta_j k(z_j, \cdot)$ to approximate the KME of original data

$\{x_i\}_{i=1}^n$, i.e. we want to solve

$$\min_{\beta,z} \left|\left| \frac{1}{n} \sum_{i=1}^n k(x_i, \cdot) - \sum_{j=1}^m \beta_j k(z_j, \cdot) \right|\right|_{\mathcal{H}} \tag{35}$$

where $\beta_j \in \mathbb{R}$ is the coefficient and $z_i \in X$ is the reduced sample. The above problem is known as the reduced set construction (Scholkopf et al., 1999), when $z_j$ is newly constructed samples. Several algorithms can be used for handling the above problem.

KME $\tilde{\mu}$ enjoys a linear convergence rate $O(e^{-m})$ to empirical KME $\hat{\mu}$ when $\mathcal{H}$ is finite dimensional, which makes it a good approximation of the distribution. Meanwhile, the raw data are inaccessible to users. In this paper, we address the problem of utilizing RKME as the specification for the learnware paradigm, aiming for the efficient retrieval and organization of learnware. Consequently, both uploaders and users of learnware are required to submit the aforementioned RKME derived from their datasets, that is, the corresponding $(\beta_j, z_j)_{j=1}^m$. In this paper, the RKME reduced set is analyzed as a public specification attached to a released model. Our goal is to quantify the additional privacy risk introduced by this specification beyond the model-only baseline.

## C. Notations and Technical Overview

This appendix summarizes the notation used throughout the paper and gives a technical overview of the proof structure. The purpose is to make the decision-theoretic layer, the RKME stability layer, and the attack-specific instantiations easier to follow.

### C.1. Notation Table

**Remark on notation.** The main text uses $B$ for the membership source bit in membership inference. To avoid overloading, we denote the energy upper bound in power-posterior attacks by $B_E$, i.e., $E_f(t; V) \in [0, B_E]$.

### C.2. Technical Overview

We now summarize the proof architecture. The analysis is organized around the following chain:

$$\text{Bayes risk} \longrightarrow \text{conditional influence} \longrightarrow \text{RKHS-smoothed distinguishability} \longrightarrow \text{RKME projection stability.}$$

Each arrow corresponds to one technical layer of the paper.

**Step 1: From learnware-inversion games to Bayes value.** The starting point is the learnware-inversion game in Definition 2.1. The adversary observes either the model-only view $O_0 = (f, \nu)$ or the full-learnware view $O_1 = (f, R_f, \nu)$, and tries to infer a finite target $T$. The risk of specification is defined as the difference between the optimal gains under these two observations:

$$\Delta_{\mathcal{G}} = G_1^\star(\mathcal{G}) - G_0^\star(\mathcal{G}).$$

The first technical observation is that this operational definition is exactly a Bayes decision value. For any observation sigma-field $\mathcal{F}$, the optimal success probability is

$$\mathcal{V}_{\mathcal{G}}(\mathcal{F}) = \mathbb{E}\left[ \max_{t \in \mathcal{T}} \Pr(T = t \mid \mathcal{F}) \right].$$

Therefore

$$\Delta_{\mathcal{G}} = \mathcal{V}_{\mathcal{G}}(O_1) - \mathcal{V}_{\mathcal{G}}(O_0).$$

This turns the supremum over all adversaries into a standard statistical decision quantity. In the balanced binary case, the Bayes value has the exact total-variation form

$$\mathcal{V}_{\mathcal{G}}(O) = \frac{1}{2}\left[ 1 + \mathrm{TV}(P_{O|1}, P_{O|0}) \right].$$

This identifies learnware-inversion as a comparison between two statistical experiments: the bare-model experiment and the full-learnware experiment.

| Notation | Meaning |
|---|---|
| $\mathcal{X}$ | Feature space. |
| $\mathcal{Y}$ | Label space. |
| $\mathcal{Z} = \mathcal{X} \times \mathcal{Y}$ | Sample space. |
| $S = (X, Y)$ | A data sample. |
| $D = (S_1, \ldots, S_n)$ | Developer-side training dataset. |
| $D_X = (X_1, \ldots, X_n)$ | Feature part of the training dataset. |
| $P$ | Underlying data distribution over $\mathcal{Z}$. |
| $P_X$ | Feature marginal of $P$. |
| $n$ | Number of developer training samples. |
| $d$ | Dimension of the feature space when $\mathcal{X} \subseteq \mathbb{R}^d$. |
| $\mathcal{A}_{\mathrm{tr}}$ | Model training algorithm. |
| $f = \mathcal{A}_{\mathrm{tr}}(D)$ | Model trained from $D$. |
| $R_f$ | Specification attached to model $f$. |
| $L_f = (f, R_f)$ | Learnware consisting of the model and its specification. |
| $D_u$ | User-side task data. |
| $R_u$ | Requirement specification generated from $D_u$. |
| $\mathcal{R}_m$ | RKME specification generator with size $m$. |
| $m$ | RKME specification size, i.e., number of reduced atoms. |

*Table 1.* Basic learnware notation.

**Step 2: From Bayes value to conditional influence.** The full-learnware observation refines the model-only observation, because $O_1 = (O_0, R_f)$. Thus $\mathcal{F}_0 = \sigma(O_0) \subseteq \mathcal{F}_1 = \sigma(O_1)$. The posterior under $O_1$ is a refinement of the posterior under $O_0$, and the specification risk is the value gained by this refinement.

The local behavior of this value is described by the functional $\Phi(p) = \max_t p_t$. Its directional derivative is

$$\Phi'_p(h) = \max_{t \in \arg\max_s p_s} h_t.$$

This shows that the first-order effect of the specification vanishes away from ties in the model-only posterior. In the binary case, if $\pi_j = \Pr(T = 1 \mid O_j)$, $a = \pi_0 - \frac{1}{2}$, and $\delta = \pi_1 - \pi_0$, then

$$\Delta_{\mathcal{G}} = \mathbb{E}[|a + \delta| - |a|].$$

Hence the specification increases Bayes value when it moves the posterior across, or close to, the model-only decision boundary.

For non-asymptotic analysis, we use the conditional influence certificate. For each value of $O_0$, let $t_0(O_0)$ be a model-only Bayes action and let $P_t^{O_0} = \mathcal{L}(R_f \mid O_0, T = t)$. Then

$$\mathrm{CInf}_{\mathcal{G}}(R_f \mid O_0) = \mathbb{E}_{O_0}\left[\sum_{t \neq t_0(O_0)} \pi_t(O_0)\,\mathrm{TV}\left(P_t^{O_0}, P_{t_0(O_0)}^{O_0}\right)\right].$$

The main decision-theoretic bound is

$$0 \leq \Delta_{\mathcal{G}} \leq \mathrm{CInf}_{\mathcal{G}}(R_f \mid O_0).$$

This certificate separates two factors. The posterior weights $\pi_t(O_0)$ measure the uncertainty left after the model-only view is fixed. The total-variation term measures how much the specification can still distinguish target values under that conditioning. Thus the model and the specification are separated at the level of conditional distinguishability.

A complementary information-theoretic consequence is

$$\Delta_{\mathcal{G}} \leq \sqrt{\frac{1}{2} I(T; R_f \mid O_0)}.$$

This gives a compact interpretation: the specification contributes little when it carries little conditional information about the target beyond the model-only observation.

| Notation | Meaning |
|---|---|
| $k : \mathcal{X} \times \mathcal{X} \to \mathbb{R}$ | Positive definite kernel. |
| $\mathcal{H}$ | Reproducing kernel Hilbert space induced by $k$. |
| $\varphi(x) = k(x, \cdot)$ | Canonical feature map. |
| $\kappa$ | Uniform kernel bound, $k(x, x) \le \kappa^2$. |
| $\mu_P = \mathbb{E}_{X \sim P_X} k(X, \cdot)$ | Population kernel mean embedding of the feature marginal. |
| $\widehat{\mu}_D = n^{-1} \sum_{i=1}^{n} k(X_i, \cdot)$ | Empirical kernel mean embedding. |
| $R_f = \{(\beta_j, z_j)\}_{j=1}^{m}$ | RKME reduced set. |
| $\widetilde{\mu}_{R_f} = \sum_{j=1}^{m} \beta_j k(z_j, \cdot)$ | Reduced mean element induced by $R_f$. |
| $\widetilde{\mu}_D$ | Reduced mean element returned by applying the RKME generator to dataset $D$. |
| $\theta = (\beta_i, z_i)$ | Parameter vector of a reduced set. |
| $\Theta_m$ | Local parameter space of reduced sets of size $m$. |
| $\Phi_m(\theta) = \sum_{j=1}^{m} \beta_j k(z_j, \cdot)$ | Parameter-to-RKHS map for reduced means. |
| $\mathcal{M}_m = \{\Phi_m(\theta) : \theta \in \Theta_m\}$ | Reduced-set family in RKHS. |
| $J_D(\theta) = \frac{1}{2} \|\widehat{\mu}_D - \Phi_m(\theta)\|_{\mathcal{H}}^2$ | Empirical RKME objective. |
| $J_\mu(\theta) = \frac{1}{2} \|\mu - \Phi_m(\theta)\|_{\mathcal{H}}^2$ | RKME objective with a generic input mean element $\mu$. |
| $\theta_D$ | Returned local RKME solution for dataset $D$. |
| $r_D = \widehat{\mu}_D - \Phi_m(\theta_D)$ | RKME residual at the returned solution. |
| $\mathcal{T}_m$ | Local RKME projection functional, $\mathcal{T}_m(\mu) = \Phi_m(\theta(\mu))$. |
| $\eta_m(D) = \|\widetilde{\mu}_D - \widehat{\mu}_D\|_{\mathcal{H}}$ | Empirical reduced-set approximation error. |
| $\mathrm{MMD}(P, Q)$ | Maximum mean discrepancy, $\|\mu_P - \mu_Q\|_{\mathcal{H}}$. |

*Table 2.* RKME and RKHS notation.

**Step 3: From conditional influence to RKHS-smoothed distinguishability.** The conditional influence bound involves total variation between conditional laws of the specification. For deterministic specifications, ordinary total variation can be too strong: two different reduced sets may give disjoint point masses even when they are close in RKHS norm. RKME is also not used in learnware through arbitrary measurable tests; it is used through kernel scores, mean-embedding distances, RKHS statistics, and posterior priors derived from such statistics.

We therefore introduce an RKHS-smoothed total variation. For laws $\mathfrak{Q}$ and $\mathfrak{Q}'$ over RKHS mean elements,

$$\mathrm{TV}_{\mathcal{H}, \Lambda}(\mathfrak{Q}, \mathfrak{Q}') = \frac{1}{2} \sup_{g \in \mathcal{G}_\Lambda} |\mathbb{E}_{\mathfrak{Q}} g(U) - \mathbb{E}_{\mathfrak{Q}'} g(U)|,$$

where $g(U) = \psi(\langle h, U \rangle_{\mathcal{H}})$, $\|h\|_{\mathcal{H}} \le \Lambda$, and $\psi$ is 1-Lipschitz and bounded in $[-1, 1]$. This is a restricted integral probability metric; we call it RKHS-smoothed total variation because it replaces arbitrary measurable tests in total variation by RKHS-compatible tests.

The bridge theorem states that

$$\mathrm{TV}_{\mathcal{H}, \Lambda}(\mathfrak{Q}, \mathfrak{Q}') \le \frac{\Lambda}{2} W_1^{\mathcal{H}}(\mathfrak{Q}, \mathfrak{Q}').$$

For point masses at $u, v \in \mathcal{H}$, this gives

$$\mathrm{TV}_{\mathcal{H}, \Lambda}(\delta_u, \delta_v) \le \frac{\Lambda}{2} \|u - v\|_{\mathcal{H}}.$$

Thus, once the adversary's use of the specification is restricted to the RKHS-compatible interface, controlling distinguishability reduces to controlling the RKHS perturbation of the released mean element.

**Step 4: RKME as a local projection in RKHS.** We next analyze how much the released RKME mean element can change when one training sample changes. The reduced-set family is

$$\mathcal{M}_m = \{\Phi_m(\theta) : \theta \in \Theta_m\} \subseteq \mathcal{H},$$

and the RKME objective is

$$J_\mu(\theta) = \frac{1}{2} \|\mu - \Phi_m(\theta)\|_{\mathcal{H}}^2.$$

Thus RKME can be viewed locally as projecting the empirical KME $\widehat{\mu}_D$ onto $\mathcal{M}_m$.

| Notation | Meaning |
|----------|---------|
| $\mathcal{G} = (\mathcal{T}, \mathsf{C})$ | A finite-target learnware-inversion game. |
| $\mathcal{T}$ | Finite target space of the privacy game. |
| $\mathsf{C}$ | Challenge kernel generating the target and side information. |
| $T$ | Hidden privacy target. |
| $\nu$ | Side information revealed by the challenger. |
| $O_0 = (f, \nu)$ | Model-only observation. |
| $O_1 = (f, R_f, \nu)$ | Full-learnware observation. |
| $\mathcal{F}_j = \sigma(O_j)$ | Sigma-field generated by observation $O_j$, $j \in \{0, 1\}$. |
| $A_j$ | Randomized adversary observing $O_j$. |
| $\widehat{T}_j$ | Prediction of $T$ made by $A_j$. |
| $\mathrm{gain}_{\mathcal{G}}(A_j; O_j)$ | Success probability of adversary $A_j$ in game $\mathcal{G}$. |
| $G_j^{\star}(\mathcal{G})$ | Optimal gain under observation $O_j$. |
| $\Delta_{\mathcal{G}}$ | Specification risk in game $\mathcal{G}$, equal to $\mathrm{Risk}_{\mathcal{G}}(R_f \mid f)$. |
| $\mathfrak{G}$ | A prescribed family of privacy games. |
| $\eta_{\mathcal{F}}(t) = \Pr(T = t \mid \mathcal{F})$ | Posterior vector under observation sigma-field $\mathcal{F}$. |
| $\Phi(p) = \max_t p_t$ | Bayes decision functional on the probability simplex. |
| $\mathcal{V}_{\mathcal{G}}(\mathcal{F})$ | Bayes value of observing $\mathcal{F}$. |
| $P_{O\mid t} = \mathcal{L}(O \mid T = t)$ | Conditional law of an observation given target value $t$. |
| $\pi_t(O_0) = \Pr(T = t \mid O_0)$ | Model-only posterior probability of target $t$. |
| $t_0(O_0)$ | A measurable model-only Bayes action. |
| $P_t^{O_0} = \mathcal{L}(R_f \mid O_0, T = t)$ | Conditional law of the specification after fixing $O_0$ and $T = t$. |
| $\mathrm{CInf}_{\mathcal{G}}(R_f \mid O_0)$ | Conditional influence certificate for the specification. |
| $I(T; R_f \mid O_0)$ | Conditional mutual information between target and specification given the model-only view. |

*Table 3.* Decision-theoretic notation.

Let $\theta_D$ be the returned local solution and $r_D = \widehat{\mu}_D - \Phi_m(\theta_D)$ be the residual. The Hessian of the RKME objective at a stationary point has the form

$$\nabla^2 J_{\widehat{\mu}_D}(\theta_D)[u, u] = \|D\Phi_m(\theta_D)[u]\|_{\mathcal{H}}^2 - \langle r_D, D^2\Phi_m(\theta_D)[u, u]\rangle_{\mathcal{H}}.$$

This formula separates the tangent Gram term from the curvature-residual correction. The curvature-residual margin is

$$\chi_D = s_D^2 - \rho_D M_D,$$

where $s_D$ measures tangent nondegeneracy, $\rho_D = \|r_D\|_{\mathcal{H}}$ is the residual size, and $M_D$ controls local curvature. When $\chi_D > 0$, the returned RKME solution is regular and the Hessian is positive definite.

The implicit function theorem then yields a local projection functional $\mathcal{T}_m$ with derivative

$$D\mathcal{T}_m(\widehat{\mu}_D)[h] = D\Phi_m(\theta_D)H_D^{-1}D\Phi_m(\theta_D)^*h.$$

Its operator norm is bounded by

$$C_{\mathrm{stab}}(D, m) = \frac{\|D\Phi_m(\theta_D)\|_{\mathrm{op}}^2}{\chi_D}.$$

Therefore, for neighboring datasets $D, D'$ that remain in the same regularity neighborhood,

$$\|\widetilde{\mu}_D - \widetilde{\mu}_{D'}\|_{\mathcal{H}} \leq \frac{2\kappa C_{\mathrm{stab}}(D, m)}{n} + o(n^{-1}).$$

This is the record-level stability statement. One sample changes the empirical KME by at most $2\kappa/n$, and the RKME generator amplifies this perturbation by its local stability factor.

The same argument gives an RKME influence function. At a regular population projection,

$$\widetilde{\mu}_D - \widetilde{\mu}_P = \frac{1}{n}\sum_{i=1}^{n} \mathrm{IF}_m(X_i; P) + o_p(n^{-1/2}),$$

where

$$\mathrm{IF}_m(x; P) = D\mathcal{T}_m(\mu_P)[k(x, \cdot) - \mu_P].$$

This expansion gives a statistical interpretation of the stability bound: each sample contributes to the released RKME through a $1/n$-scaled influence function.

| Notation | Meaning |
|---|---|
| $D\Phi_m(\theta)$ | First derivative of the reduced-set map. |
| $D^2\Phi_m(\theta)$ | Second derivative of the reduced-set map. |
| $H_D = \nabla^2 J_{\widehat{\mu}_D}(\theta_D)$ | Hessian of the RKME objective at the returned solution. |
| $s_D^2 = \inf_{\|u\|=1}\|D\Phi_m(\theta_D)[u]\|_{\mathcal{H}}^2$ | Tangent nondegeneracy term. |
| $\rho_D = \|r_D\|_{\mathcal{H}}$ | RKME residual norm. |
| $M_D = \sup_{\|u\|=1}\|D^2\Phi_m(\theta_D)[u,u]\|_{\mathcal{H}}$ | Local curvature bound of the reduced-set map. |
| $\chi_D = s_D^2 - \rho_D M_D$ | Curvature-residual margin. |
| $C_{\mathrm{stab}}(D,m)$ | Local RKME projection-stability factor. |
| $C_{\mathrm{stab}}(m)$ | Uniform upper bound of $C_{\mathrm{stab}}(D,m)$ over relevant datasets. |
| $\mathrm{IF}_m(x;P)$ | RKME influence function at sample $x$. |
| $\mathcal{G}_\Lambda$ | RKHS-compatible test class with RKHS norm budget $\Lambda$. |
| $\mathrm{TV}_{\mathcal{H},\Lambda}$ | RKHS-smoothed total variation over $\mathcal{G}_\Lambda$. |
| $W_1^{\mathcal{H}}$ | Wasserstein-1 distance induced by the RKHS norm. |
| $\Lambda$ | RKHS norm budget of specification-dependent tests. |
| $B$ | Membership source bit in membership inference games. |
| $S^\star$ | Challenge sample in membership or attribute inference. |
| $a : \mathcal{Z} \to \mathcal{A}$ | Sensitive attribute map. |
| $b : \mathcal{Z} \to \mathcal{B}$ | Non-sensitive side-information map. |
| $K = |\mathcal{A}|$ | Number of possible sensitive attribute values. |
| $c_0$ | Normalization margin for RKME-linear attribute priors. |
| $V$ | Context available to a posterior-style adversary. |
| $E_f(t;V)$ | Model-induced energy for target value $t$. |
| $\rho_R(t \mid V)$ | Specification-induced prior over target values. |
| $\rho_0(t \mid V)$ | Model-only baseline prior over target values. |
| $q_R^\gamma(t \mid V)$ | Power posterior using the RKME specification. |
| $\gamma$ | Temperature parameter of the power posterior. |
| $B_E$ | Uniform upper bound for the model energy, $E_f(t;V) \in [0, B_E]$. |
| $m_* = \lfloor\sqrt{n}\rfloor$ | Square-root RKME size used in the trade-off analysis. |
| $C_0, \xi$ | Constants in the subquadratic stability condition $C_{\mathrm{stab}}(m) \leq C_0 m^{2-\xi}$. |

*Table 4.* RKME stability and attack-instantiation notation.

**Step 5: RKME control of specification risk.** Combining the previous steps gives the main risk-control chain. For RKHS-compatible adversaries,

$$\Delta_{\mathcal{G}}^{\mathcal{H},\Lambda} \leq \mathrm{CInf}_{\mathcal{G}}^{\mathcal{H},\Lambda}.$$

The RKHS-smoothed TV bridge controls the conditional distinguishability of RKME mean elements by their RKHS distance. The local RKME projection-stability theorem then controls this RKHS distance by $C_{\mathrm{stab}}(m)/n$. Thus, for games whose relevant conditional datasets can be coupled by replacing one record,

$$\Delta_{\mathcal{G}}^{\mathcal{H},\Lambda} \leq \frac{\Lambda\kappa C_{\mathrm{stab}}(m)}{n} + o(n^{-1}),$$

up to attack-dependent constants.

This is the central theorem behind all later instantiations. The statement should be read as a specification-side guarantee. The model-only information is already included in $O_0$; the bound controls the additional value of the RKME specification under the RKHS-compatible interface.

**Step 6: Attack instantiations.** The attack-specific results are direct specializations of the general bound.

For loss-based membership inference, the target is the source bit in the membership challenge, and the model-only view includes the model loss on the challenge sample. The RKME specification can only help through an additional RKHS-compatible statistic. This gives

$$\Delta_{\text{loss-MIA}}^{\mathcal{H},\Lambda} \leq C_{\mathrm{loss}}\frac{\Lambda\kappa C_{\mathrm{stab}}(m)}{n} + o(n^{-1}).$$

The constant $C_{\mathrm{loss}}$ captures the Lipschitz dependence of the final decision rule on the specification-dependent statistic.

For posterior-style attacks, the model supplies an energy $E_f(t; V)$ and the specification supplies a prior $\rho_R(t \mid V)$. The power posterior is

$$q_R^\gamma(t \mid V) \propto \rho_R(t \mid V) \exp\{-E_f(t; V)/\gamma\}.$$

If $E_f(t; V) \in [0, B_E]$, then weighted normalization gives

$$\mathrm{TV}(q_R^\gamma, q_0^\gamma) \leq e^{B_E/\gamma} \, \mathrm{TV}(\rho_R, \rho_0).$$

Thus the posterior attack is controlled once the specification-induced prior shift is controlled. This yields a risk bound of order

$$e^{B_E/\gamma} \frac{\Lambda \kappa C_{\mathrm{stab}}(m)}{n}.$$

For finite attribute inference, the target is a finite sensitive attribute $A = a(S^\star)$. If the specification-induced prior is constructed from RKME-linear scores and the normalization denominator is bounded below by $c_0$, then the normalization map is Lipschitz. This gives

$$\Delta_{\mathrm{AIA}}^{\mathcal{H},\Lambda} \leq \frac{2K\Lambda \kappa C_{\mathrm{stab}}(m)}{c_0 n} + o(n^{-1}).$$

This controls the sample-specific incremental contribution of the RKME specification. Population-level correlations between sensitive and non-sensitive variables are interpreted separately, because they are distributional information rather than record-level instability of the released specification.

**Step 7: Search–risk trade-off.** RKME is used because it supports learnware identification. Its search utility is governed by the RKHS approximation error

$$\|\widetilde{\mu}_D - \mu_P\|_{\mathcal{H}} \leq \eta_m(D) + O_p(n^{-1/2}),$$

where $\eta_m(D) = \|\widetilde{\mu}_D - \widehat{\mu}_D\|_{\mathcal{H}}$ is the reduced-set approximation error. Increasing $m$ usually decreases $\eta_m(D)$ and improves search.

The privacy side is governed by

$$\frac{C_{\mathrm{stab}}(m)}{n}.$$

Increasing $m$ may change the geometry of $\mathcal{M}_m$ and increase $C_{\mathrm{stab}}(m)$, especially when reduced atoms become redundant or the tangent Gram matrix becomes ill-conditioned. Thus $m$ affects search through $\eta_m(D)$ and risk through $C_{\mathrm{stab}}(m)$.

The square-root specification size $m_* = \lfloor \sqrt{n} \rfloor$ is analyzed through the subquadratic projection-stability condition

$$C_{\mathrm{stab}}(m) \leq C_0 m^{2-\xi}, \qquad 1 \leq m \leq m_*,$$

for constants $C_0 > 0$ and $\xi > 0$. Under this condition,

$$\Delta_{\mathcal{G}}^{\mathcal{H},\Lambda}(m_*) \leq C_{\mathcal{G}} \Lambda \kappa C_0 m_*^{-\xi} + o(n^{-1}).$$

If additionally $\eta_m(D) = O(m^{-\alpha})$, then

$$\|\widetilde{\mu}_D - \mu_P\|_{\mathcal{H}} = O(m_*^{-\alpha}) + O_p(m_*^{-1}).$$

Thus, at the square-root size, the specification remains increasingly informative for search while the record-level incremental risk is controlled whenever the RKME projection remains subquadratically stable.

## D. Proofs for Section 3

We collect the proofs for the decision-theoretic results. Throughout this section, fix a finite-target learnware-inversion game $\mathcal{G} = (\mathcal{T}, \mathsf{C})$. We write $O_0 = (f, \nu)$, $O_1 = (f, R_f, \nu)$, $\mathcal{F}_j = \sigma(O_j)$ for $j \in \{0, 1\}$, and $\Delta_{\mathcal{G}} = \mathrm{Risk}_{\mathcal{G}}(R_f \mid f)$. Since $\mathcal{T}$ is finite and all observation spaces are standard Borel, regular conditional probabilities exist and measurable Bayes rules can be chosen by a deterministic tie-breaking rule.

### D.1. Proof of Theorem 3.1

We first prove the Bayes value identity. Let $O$ be any observation and let $\mathcal{F} = \sigma(O)$. For a randomized adversary $A$, write $A(t \mid O)$ for the conditional probability that $A$ outputs $t \in \mathcal{T}$ after observing $O$. Its gain is

$$\Pr[A(O) = T] = \mathbb{E}\left[\sum_{t \in \mathcal{T}} \mathbf{1}\{T = t\} A(t \mid O)\right].$$

Taking conditional expectation given $\mathcal{F}$ gives

$$\Pr[A(O) = T] = \mathbb{E}\left[\sum_{t \in \mathcal{T}} \Pr(T = t \mid \mathcal{F}) A(t \mid O)\right].$$

For each realized observation, $A(\cdot \mid O)$ is a probability vector over $\mathcal{T}$. Therefore

$$\sum_{t \in \mathcal{T}} \Pr(T = t \mid \mathcal{F}) A(t \mid O) \leq \max_{t \in \mathcal{T}} \Pr(T = t \mid \mathcal{F}).$$

Taking expectation yields

$$\mathrm{gain}_{\mathcal{G}}(A; O) \leq \mathbb{E}\left[\max_{t \in \mathcal{T}} \Pr(T = t \mid \mathcal{F})\right] = \mathcal{V}_{\mathcal{G}}(O).$$

This upper bound is achieved by the Bayes rule. Since $\mathcal{T}$ is finite, choose a fixed ordering on $\mathcal{T}$ and define

$$A^{\star}(O) = \min \arg \max_{t \in \mathcal{T}} \Pr(T = t \mid O),$$

where the minimum is with respect to the fixed ordering. This is measurable and satisfies

$$\mathrm{gain}_{\mathcal{G}}(A^{\star}; O) = \mathcal{V}_{\mathcal{G}}(O).$$

Applying this to $O_0$ and $O_1$ gives

$$G_j^{\star}(\mathcal{G}) = \mathcal{V}_{\mathcal{G}}(O_j), \qquad j \in \{0, 1\},$$

and hence

$$\Delta_{\mathcal{G}} = \mathcal{V}_{\mathcal{G}}(O_1) - \mathcal{V}_{\mathcal{G}}(O_0).$$

We now prove the binary total-variation identity. Assume $T \in \{0, 1\}$ and $\Pr(T = 0) = \Pr(T = 1) = 1/2$. Let $P_{O|t} = \mathcal{L}(O \mid T = t)$, and let $\lambda = P_{O|0} + P_{O|1}$ dominate both conditional laws. Write $p_t = dP_{O|t}/d\lambda$. For any deterministic classifier $a(O) \in \{0, 1\}$, its success probability is

$$\frac{1}{2} P_{O|1}(a(O) = 1) + \frac{1}{2} P_{O|0}(a(O) = 0).$$

The optimal classifier chooses class 1 whenever $p_1 \geq p_0$ and class 0 otherwise. Therefore

$$\mathcal{V}_{\mathcal{G}}(O) = \frac{1}{2} \int \max\{p_1, p_0\} \, d\lambda.$$

Using $\max\{p_1, p_0\} = (p_1 + p_0 + |p_1 - p_0|)/2$, we obtain

$$\mathcal{V}_{\mathcal{G}}(O) = \frac{1}{2}\left[1 + \frac{1}{2} \int |p_1 - p_0| \, d\lambda\right].$$

Since $\mathrm{TV}(P_{O|1}, P_{O|0}) = \frac{1}{2} \int |p_1 - p_0| \, d\lambda$, this gives

$$\mathcal{V}_{\mathcal{G}}(O) = \frac{1}{2}\left[1 + \mathrm{TV}(P_{O|1}, P_{O|0})\right].$$

This proves Theorem 3.1. $\qquad\square$

## D.2. Value monotonicity and the Blackwell interpretation

We record the value monotonicity used in the main text. Since $O_0$ is a measurable function of $O_1$, we have $\mathcal{F}_0 \subseteq \mathcal{F}_1$. Let $\eta_j(t) = \Pr(T = t \mid \mathcal{F}_j)$. By the tower property,

$$\eta_0 = \mathbb{E}[\eta_1 \mid \mathcal{F}_0].$$

The map $\Phi(p) = \max_t p_t$ is convex on the probability simplex. Hence Jensen's inequality gives

$$\Phi(\eta_0) = \Phi(\mathbb{E}[\eta_1 \mid \mathcal{F}_0]) \leq \mathbb{E}[\Phi(\eta_1) \mid \mathcal{F}_0].$$

Taking expectation,

$$\mathcal{V}_{\mathcal{G}}(O_0) \leq \mathcal{V}_{\mathcal{G}}(O_1).$$

Thus every privacy game has nonnegative specification risk. For a family $\mathfrak{G}$ of privacy games, the aggregate gap

$$\Delta_{\mathfrak{G}}(O_1, O_0) = \sup_{\mathcal{G} \in \mathfrak{G}} \{\mathcal{V}_{\mathcal{G}}(O_1) - \mathcal{V}_{\mathcal{G}}(O_0)\}$$

is therefore a one-sided value-of-information gap between the bare-model experiment and the full-learnware experiment.

## D.3. Proof of Theorem 3.2

We first prove the directional derivative of $\Phi(p) = \max_{t \in \mathcal{T}} p_t$. Fix $p$ in the probability simplex and let $M(p) = \arg\max_t p_t$. Let $h$ satisfy $\sum_t h_t = 0$. For $\epsilon > 0$,

$$\frac{\Phi(p + \epsilon h) - \Phi(p)}{\epsilon} = \max_{t \in \mathcal{T}} \left\{ h_t + \frac{p_t - \Phi(p)}{\epsilon} \right\}.$$

If $t \in M(p)$, then $p_t - \Phi(p) = 0$. If $t \notin M(p)$, then $p_t - \Phi(p) < 0$, and the term $(p_t - \Phi(p))/\epsilon$ tends to $-\infty$ as $\epsilon \downarrow 0$. Therefore

$$\lim_{\epsilon \downarrow 0} \frac{\Phi(p + \epsilon h) - \Phi(p)}{\epsilon} = \max_{t \in M(p)} h_t.$$

This proves Hadamard directional differentiability of $\Phi$ in finite dimension.

Now set $\eta_j(t) = \Pr(T = t \mid \mathcal{F}_j)$ and $H = \eta_1 - \eta_0$. Since $\eta_0 = \mathbb{E}[\eta_1 \mid \mathcal{F}_0]$, we have $\mathbb{E}[H \mid \mathcal{F}_0] = 0$. For every $\epsilon \in [0, 1]$, $\eta_0 + \epsilon H$ is bounded in $\mathbb{R}^{|\mathcal{T}|}$, and

$$\left| \frac{\Phi(\eta_0 + \epsilon H) - \Phi(\eta_0)}{\epsilon} \right| \leq \|H\|_\infty \leq 1.$$

Dominated convergence gives

$$\frac{d}{d\epsilon}\bigg|_{\epsilon=0+} \mathbb{E}[\Phi(\eta_0 + \epsilon H)] = \mathbb{E}\left[ \max_{t \in M(\eta_0)} H_t \right].$$

If the model-only posterior has a unique maximizer almost surely, write it as $t^\star(O_0)$. Then $t^\star$ is $\mathcal{F}_0$-measurable and

$$\mathbb{E}[H_{t^\star}] = \mathbb{E}[\mathbb{E}[H_{t^\star} \mid \mathcal{F}_0]] = 0.$$

Hence the first-order derivative is zero.

We now prove the binary representation. Let $T \in \{0, 1\}$ and define $\pi_j = \Pr(T = 1 \mid \mathcal{F}_j)$, $a = \pi_0 - \frac{1}{2}$, and $\delta = \pi_1 - \pi_0$. For a binary target,

$$\Phi(\eta_j) = \max\{\pi_j, 1 - \pi_j\} = \frac{1}{2} + \left| \pi_j - \frac{1}{2} \right|.$$

Therefore

$$\Delta_{\mathcal{G}} = \mathbb{E}[\Phi(\eta_1) - \Phi(\eta_0)] = \mathbb{E}[|a + \delta| - |a|].$$

It remains to compute the conditional contribution. Since $\mathbb{E}[\delta \mid \mathcal{F}_0] = 0$, fix $\mathcal{F}_0$ and treat $a$ as constant. If $a > 0$, then

$$|a + \delta| - a = \delta - 2(a + \delta)\mathbf{1}\{a + \delta < 0\}.$$

Taking conditional expectation and using $\mathbb{E}[\delta \mid \mathcal{F}_0] = 0$,

$$\mathbb{E}[|a + \delta| - |a| \mid \mathcal{F}_0] = 2\,\mathbb{E}[(-(a + \delta))_+ \mid \mathcal{F}_0].$$

If $a < 0$, then

$$|a + \delta| - |a| = |a + \delta| + a,$$

and similarly

$$\mathbb{E}[|a + \delta| - |a| \mid \mathcal{F}_0] = 2\,\mathbb{E}[(a + \delta)_+ \mid \mathcal{F}_0].$$

Both cases are summarized by

$$2\,\mathbb{E}\left[\left(-\operatorname{sgn}(a)(a + \delta)\right)_+ \mid \mathcal{F}_0\right] \quad \text{on } \{a \neq 0\}.$$

On the boundary $a = 0$, the contribution is directly

$$\mathbb{E}[|\delta| \mid \mathcal{F}_0].$$

This proves Theorem 3.2. $\qquad\qquad\square$

## D.4. Proof of Theorem 3.4

We prove the general finite-target statement. Fix a realized value $O_0 = o$. Write $\pi_t = \Pr(T = t \mid O_0 = o)$ and choose $t_0 \in \arg\max_t \pi_t$. Let $P_t = \mathcal{L}(R_f \mid O_0 = o, T = t)$. Let $\lambda = \sum_{t \in \mathcal{T}} P_t$ dominate all $P_t$, and write $p_t = dP_t/d\lambda$.

Given $O_0 = o$, the model-only Bayes value is $\max_t \pi_t = \pi_{t_0}$. After observing $R_f$, the conditional law of $R_f$ is $M = \sum_t \pi_t P_t$, and the full Bayes value conditional on $O_0 = o$ is

$$\int \max_{t \in \mathcal{T}} \pi_t p_t(r)\, d\lambda(r).$$

Thus the conditional gain from the specification is

$$\int \left[\max_t \pi_t p_t - \pi_{t_0} p_{t_0}\right] d\lambda.$$

For nonnegative numbers $\{a_t\}_{t \in \mathcal{T}}$,

$$\max_t a_t - a_{t_0} \leq \sum_{t \neq t_0} (a_t - a_{t_0})_+.$$

Applying this with $a_t = \pi_t p_t$ gives

$$\max_t \pi_t p_t - \pi_{t_0} p_{t_0} \leq \sum_{t \neq t_0} (\pi_t p_t - \pi_{t_0} p_{t_0})_+.$$

Since $t_0$ is a maximizer, $\pi_t \leq \pi_{t_0}$ for every $t \neq t_0$. Therefore

$$\pi_t p_t - \pi_{t_0} p_{t_0} \leq \pi_t (p_t - p_{t_0}),$$

and hence

$$(\pi_t p_t - \pi_{t_0} p_{t_0})_+ \leq \pi_t (p_t - p_{t_0})_+.$$

Integrating,

$$\int (\pi_t p_t - \pi_{t_0} p_{t_0})_+ d\lambda \leq \pi_t \int (p_t - p_{t_0})_+ d\lambda.$$

For probability measures, $\int (p_t - p_{t_0})_+ \, d\lambda = \mathrm{TV}(P_t, P_{t_0})$. Therefore the conditional gain is bounded by

$$\sum_{t \neq t_0} \pi_t \,\mathrm{TV}(P_t, P_{t_0}).$$

Taking expectation over $O_0$ gives

$$\Delta_{\mathcal{G}} \leq \mathrm{CInf}_{\mathcal{G}}(R_f \mid O_0).$$

The lower bound $\Delta_{\mathcal{G}} \geq 0$ follows from the value monotonicity proved above.

For the binary case, write $\pi(O_0) = \Pr(T = 1 \mid O_0)$. If $\pi(O_0) \geq 1/2$, then $t_0 = 1$ and the coefficient of the only competing target is $1 - \pi(O_0)$. If $\pi(O_0) < 1/2$, then $t_0 = 0$ and the coefficient is $\pi(O_0)$. Hence

$$\mathrm{CInf}_{\mathcal{G}}(R_f \mid O_0) = \mathbb{E}_{O_0}\left[\min\{\pi(O_0), 1 - \pi(O_0)\}\,\mathrm{TV}(P_1^{O_0}, P_0^{O_0})\right],$$

where $P_t^{O_0} = \mathcal{L}(R_f \mid O_0, T = t)$. This proves Theorem 3.4. $\qquad\square$

### D.5. Proof of Corollary 3.5

Let $\eta_j = \mathcal{L}(T \mid O_j)$ be the posterior distribution over $\mathcal{T}$ under observation $O_j$. The Bayes value can be written as $\mathbb{E}[\Phi(\eta_j)]$, where $\Phi(p) = \max_t p_t$. For any two probability vectors $p, q$ on a finite set,

$$|\Phi(p) - \Phi(q)| \leq \|p - q\|_\infty \leq \mathrm{TV}(p, q),$$

where for discrete distributions $\mathrm{TV}(p, q) = \frac{1}{2}\|p - q\|_1$ because $p - q$ has total mass zero. Therefore

$$\Delta_{\mathcal{G}} = \mathbb{E}[\Phi(\eta_1) - \Phi(\eta_0)] \leq \mathbb{E}[\mathrm{TV}(\eta_1, \eta_0)].$$

This gives Eq. (12).

For the mutual-information bound, use Pinsker's inequality conditionally:

$$\mathrm{TV}\left(\mathcal{L}(T \mid O_1), \mathcal{L}(T \mid O_0)\right) \leq \sqrt{\frac{1}{2}D_{\mathrm{KL}}\left(\mathcal{L}(T \mid O_1) \,\|\, \mathcal{L}(T \mid O_0)\right)}.$$

Taking expectation and applying Jensen's inequality to the concave function $\sqrt{\cdot}$,

$$\mathbb{E}[\mathrm{TV}(\mathcal{L}(T \mid O_1), \mathcal{L}(T \mid O_0))] \leq \sqrt{\frac{1}{2}\mathbb{E}D_{\mathrm{KL}}\left(\mathcal{L}(T \mid O_1) \,\|\, \mathcal{L}(T \mid O_0)\right)}.$$

The expectation of this conditional KL divergence is precisely the conditional mutual information $I(T; R_f \mid O_0)$, because $O_1 = (O_0, R_f)$. Therefore

$$\Delta_{\mathcal{G}} \leq \sqrt{\frac{1}{2}I(T; R_f \mid O_0)}.$$

This proves Corollary 3.5. $\qquad\square$

## E. Proofs for Section 4

We now prove the RKME projection-stability results. We work in a finite-dimensional local coordinate chart of the reduced-set parameter space. If the implementation uses constraints on weights or reduced points, the same proof applies in any smooth local chart around an interior regular solution, or on the corresponding active smooth manifold. We write $\Phi = \Phi_m$ when $m$ is fixed.

Throughout this section, $\Theta_m$ denotes the local parameter space, $\Phi : \Theta_m \to \mathcal{H}$ is twice continuously differentiable, and

$$J_\mu(\theta) = \frac{1}{2}\|\mu - \Phi(\theta)\|_{\mathcal{H}}^2.$$

For a dataset $D$, $\mu = \widehat{\mu}_D$, $\theta_D$ is the returned local solution, $\widetilde{\mu}_D = \Phi(\theta_D)$, and $r_D = \widehat{\mu}_D - \Phi(\theta_D)$.

### E.1. Proof of Theorem 4.2

Let $u$ be a parameter direction. Define the curve $\theta(t) = \theta_D + tu$ and

$$r(t) = \widehat{\mu}_D - \Phi(\theta(t)).$$

Then $J_{\widehat{\mu}_D}(\theta(t)) = \frac{1}{2}\|r(t)\|_{\mathcal{H}}^2$. Differentiating once gives

$$\frac{d}{dt}J_{\widehat{\mu}_D}(\theta(t)) = \langle r(t), r'(t)\rangle_{\mathcal{H}} = -\langle r(t), D\Phi(\theta(t))[u]\rangle_{\mathcal{H}}.$$

At a stationary point $\theta_D$, this derivative vanishes for all $u$, equivalently

$$\langle r_D, D\Phi(\theta_D)[u]\rangle_{\mathcal{H}} = 0, \qquad \forall u.$$

Differentiating a second time at $t = 0$,

$$\frac{d^2}{dt^2}J_{\widehat{\mu}_D}(\theta(t))\bigg|_{t=0} = \|D\Phi(\theta_D)[u]\|_{\mathcal{H}}^2 - \langle r_D, D^2\Phi(\theta_D)[u,u]\rangle_{\mathcal{H}}.$$

This is Eq. (15).

By the definition of $s_D$, $M_D$, and $\rho_D$,

$$\|D\Phi(\theta_D)[u]\|_{\mathcal{H}}^2 \geq s_D^2\|u\|^2$$

and

$$\left|\langle r_D, D^2\Phi(\theta_D)[u,u]\rangle_{\mathcal{H}}\right| \leq \rho_D M_D\|u\|^2.$$

Therefore

$$\nabla^2 J_{\widehat{\mu}_D}(\theta_D)[u,u] \geq (s_D^2 - \rho_D M_D)\|u\|^2 = \chi_D\|u\|^2.$$

If $\chi_D > 0$, then $\nabla^2 J_{\widehat{\mu}_D}(\theta_D) \succeq \chi_D I$. This proves Theorem 4.2. $\qquad\square$

### E.2. Proof of Theorem 4.3

Define the first-order optimality map

$$F(\theta, \mu) = D\Phi(\theta)^*\big(\Phi(\theta) - \mu\big),$$

where $D\Phi(\theta)^* : \mathcal{H} \to T_\theta^*\Theta_m$ is the adjoint of the derivative. A stationary point of $J_\mu$ is exactly a solution of $F(\theta, \mu) = 0$.

At $(\theta_D, \widehat{\mu}_D)$, the derivative of $F$ with respect to $\theta$ is the Hessian

$$D_\theta F(\theta_D, \widehat{\mu}_D) = H_D = \nabla^2 J_{\widehat{\mu}_D}(\theta_D).$$

By regularity, $H_D \succeq \chi_D I$ and hence $H_D$ is invertible with $\|H_D^{-1}\|_{\text{op}} \leq 1/\chi_D$. The Banach-space implicit function theorem therefore gives neighborhoods $\mathcal{U}$ of $\widehat{\mu}_D$ and $\mathcal{V}$ of $\theta_D$, together with a continuously differentiable map $\theta(\mu) : \mathcal{U} \to \mathcal{V}$ such that $F(\theta(\mu), \mu) = 0$ and $\theta(\widehat{\mu}_D) = \theta_D$.

To compute the derivative, differentiate the identity $F(\theta(\mu), \mu) = 0$ in the direction $h \in \mathcal{H}$:

$$H_D\, D\theta_{\widehat{\mu}_D}[h] - D\Phi(\theta_D)^* h = 0.$$

Thus

$$D\theta_{\widehat{\mu}_D}[h] = H_D^{-1}D\Phi(\theta_D)^* h.$$

Since $\mathcal{T}_m(\mu) = \Phi(\theta(\mu))$, the chain rule gives

$$D\mathcal{T}_m(\widehat{\mu}_D)[h] = D\Phi(\theta_D)H_D^{-1}D\Phi(\theta_D)^* h.$$

Consequently,

$$\|D\mathcal{T}_m(\widehat{\mu}_D)\|_{\text{op}} \leq \|D\Phi(\theta_D)\|_{\text{op}}\|H_D^{-1}\|_{\text{op}}\|D\Phi(\theta_D)^*\|_{\text{op}}$$
$$= \frac{\|D\Phi(\theta_D)\|_{\text{op}}^2}{\chi_D} = C_{\text{stab}}(D, m).$$

Since $\mathcal{T}_m$ is differentiable at $\widehat{\mu}_D$,

$$\mathcal{T}_m(\mu') - \mathcal{T}_m(\mu) = D\mathcal{T}_m(\mu)[\mu' - \mu] + o(\|\mu' - \mu\|_{\mathcal{H}})$$

as $\mu' \to \mu$ within the local neighborhood. This yields

$$\|\mathcal{T}_m(\mu') - \mathcal{T}_m(\mu)\|_{\mathcal{H}} \leq C_{\text{stab}}(D, m)\|\mu' - \mu\|_{\mathcal{H}} + o(\|\mu' - \mu\|_{\mathcal{H}}).$$

Finally, suppose $D$ and $D'$ differ in one feature record, say $X_i$ is replaced by $X_i'$. Then

$$\widehat{\mu}_D - \widehat{\mu}_{D'} = \frac{1}{n}\big(k(X_i, \cdot) - k(X_i', \cdot)\big).$$

Since $k(x, x) \leq \kappa^2$, $\|k(x, \cdot)\|_{\mathcal{H}} \leq \kappa$, and therefore

$$\|\widehat{\mu}_D - \widehat{\mu}_{D'}\|_{\mathcal{H}} \leq \frac{2\kappa}{n}.$$

If $\widehat{\mu}_{D'}$ lies in the local neighborhood of $\widehat{\mu}_D$; in particular, for all sufficiently large $n$ whenever the neighborhood radius is fixed, the previous local estimate gives

$$\|\widetilde{\mu}_D - \widetilde{\mu}_{D'}\|_{\mathcal{H}} \leq \frac{2\kappa C_{\text{stab}}(D, m)}{n} + o(n^{-1}).$$

This proves Theorem 4.3. □

### E.3. Proof of Theorem 4.4

Let $\mu_P = \mathbb{E}k(X, \cdot)$ be the population KME. Under the regularity condition at the population projection, Theorem 4.3 applied at $\mu_P$ shows that $\mathcal{T}_m$ is continuously differentiable, hence Hadamard differentiable, at $\mu_P$. Its derivative is

$$D\mathcal{T}_m(\mu_P)[h] = D\Phi(\theta_P)H_P^{-1}D\Phi(\theta_P)^*h,$$

where $H_P = \nabla^2 J_{\mu_P}(\theta_P)$.

The empirical KME satisfies

$$\widehat{\mu}_D - \mu_P = \frac{1}{n}\sum_{i=1}^{n}\big(k(X_i, \cdot) - \mu_P\big).$$

Since $k(x, x) \leq \kappa^2$, the summands are bounded in $\mathcal{H}$ by $2\kappa$, and hence $\|\widehat{\mu}_D - \mu_P\|_{\mathcal{H}} = O_p(n^{-1/2})$. By Hadamard differentiability,

$$\mathcal{T}_m(\widehat{\mu}_D) - \mathcal{T}_m(\mu_P) = D\mathcal{T}_m(\mu_P)[\widehat{\mu}_D - \mu_P] + o_p(n^{-1/2}).$$

Substituting the empirical average gives

$$\widetilde{\mu}_D - \widetilde{\mu}_P = \frac{1}{n}\sum_{i=1}^{n}D\mathcal{T}_m(\mu_P)[k(X_i, \cdot) - \mu_P] + o_p(n^{-1/2})$$

$$= \frac{1}{n}\sum_{i=1}^{n}\text{IF}_m(X_i; P) + o_p(n^{-1/2}),$$

where

$$\text{IF}_m(x; P) = D\mathcal{T}_m(\mu_P)[k(x, \cdot) - \mu_P].$$

Using the derivative formula,

$$\text{IF}_m(x; P) = D\Phi(\theta_P)H_P^{-1}D\Phi(\theta_P)^*[k(x, \cdot) - \mu_P].$$

Moreover,

$$\|k(x, \cdot) - \mu_P\|_{\mathcal{H}} \leq \|k(x, \cdot)\|_{\mathcal{H}} + \|\mu_P\|_{\mathcal{H}} \leq 2\kappa,$$

and $\|D\mathcal{T}_m(\mu_P)\|_{\text{op}} \leq C_{\text{stab}}(P, m)$. Hence

$$\|\text{IF}_m(x; P)\|_{\mathcal{H}} \leq 2\kappa C_{\text{stab}}(P, m).$$

This proves Theorem 4.4. □

### E.4. Proof of Theorem 4.5

Let $\mathfrak{Q}$ and $\mathfrak{Q}'$ be probability laws over RKHS mean elements. Recall that

$$\mathrm{TV}_{\mathcal{H},\Lambda}(\mathfrak{Q}, \mathfrak{Q}') = \frac{1}{2} \sup_{g \in \mathcal{G}_\Lambda} |\mathbb{E}_{\mathfrak{Q}} g(U) - \mathbb{E}_{\mathfrak{Q}'} g(V)|,$$

where $g(U) = \psi(\langle h, U \rangle_{\mathcal{H}})$, $\|h\|_{\mathcal{H}} \leq \Lambda$, and $\psi : \mathbb{R} \to [-1, 1]$ is 1-Lipschitz.

Let $\Gamma$ be any coupling of $\mathfrak{Q}$ and $\mathfrak{Q}'$. For $g \in \mathcal{G}_\Lambda$,

$$
\begin{aligned}
|\mathbb{E}_{\mathfrak{Q}} g(U) - \mathbb{E}_{\mathfrak{Q}'} g(V)| &= |\mathbb{E}_\Gamma [g(U) - g(V)]| \\
&\leq \mathbb{E}_\Gamma [|\psi(\langle h, U \rangle) - \psi(\langle h, V \rangle)|] \\
&\leq \mathbb{E}_\Gamma [|\langle h, U - V \rangle_{\mathcal{H}}|] \\
&\leq \Lambda \mathbb{E}_\Gamma \|U - V\|_{\mathcal{H}}.
\end{aligned}
$$

Taking the supremum over $g \in \mathcal{G}_\Lambda$ and then the infimum over all couplings $\Gamma$ gives

$$\mathrm{TV}_{\mathcal{H},\Lambda}(\mathfrak{Q}, \mathfrak{Q}') \leq \frac{\Lambda}{2} W_1^{\mathcal{H}}(\mathfrak{Q}, \mathfrak{Q}').$$

If $\mathfrak{Q} = \delta_u$ and $\mathfrak{Q}' = \delta_v$, then $W_1^{\mathcal{H}}(\delta_u, \delta_v) = \|u - v\|_{\mathcal{H}}$, so

$$\mathrm{TV}_{\mathcal{H},\Lambda}(\delta_u, \delta_v) \leq \frac{\Lambda}{2} \|u - v\|_{\mathcal{H}}.$$

If $u = \mu_P$ and $v = \mu_Q$, then $\|\mu_P - \mu_Q\|_{\mathcal{H}} = \mathrm{MMD}(P, Q)$, yielding the final statement. $\qquad\square$

### E.5. Proof of Theorem 4.6

The proof follows the same conditioning argument as Theorem 3.4, with ordinary total variation replaced by the RKHS-smoothed integral probability metric induced by $\mathcal{G}_\Lambda$.

Fix $O_0 = o$. Let $\pi_t = \Pr(T = t \mid O_0 = o)$ and let $t_0 \in \arg\max_t \pi_t$. Let $\mathfrak{P}_t^o$ be the conditional law of the released RKME mean element $\widetilde{\mu}_D$ given $(O_0 = o, T = t)$. For an RKHS-compatible adversary, each specification-dependent comparison between target $t$ and the baseline target $t_0$ is represented by a bounded test in $\mathcal{G}_\Lambda$. Hence the same argument as in the proof of Theorem 3.4 gives the conditional bound

$$\text{conditional gain at } o \leq \sum_{t \neq t_0} \pi_t \, \mathrm{TV}_{\mathcal{H},\Lambda}(\mathfrak{P}_t^o, \mathfrak{P}_{t_0}^o).$$

Taking expectation over $O_0$ gives

$$\Delta_{\mathcal{G}}^{\mathcal{H},\Lambda} \leq \mathrm{CInf}_{\mathcal{G}}^{\mathcal{H},\Lambda}.$$

Now assume the coupling condition in the theorem. For each $o$ and each relevant pair $(t, t_0)$, couple the conditional datasets so that the two datasets differ in one record. Under the uniform regularity assumption, Theorem 4.3 gives

$$\|\widetilde{\mu}_D - \widetilde{\mu}_{D'}\|_{\mathcal{H}} \leq \frac{2\kappa C_{\mathrm{stab}}(m)}{n} + o(n^{-1})$$

along the coupling. Therefore

$$W_1^{\mathcal{H}}(\mathfrak{P}_t^o, \mathfrak{P}_{t_0}^o) \leq \frac{2\kappa C_{\mathrm{stab}}(m)}{n} + o(n^{-1}).$$

By Theorem 4.5,

$$\mathrm{TV}_{\mathcal{H},\Lambda}(\mathfrak{P}_t^o, \mathfrak{P}_{t_0}^o) \leq \frac{\Lambda \kappa C_{\mathrm{stab}}(m)}{n} + o(n^{-1}).$$

Since $\sum_{t \neq t_0} \pi_t \leq 1$, we obtain

$$\mathrm{CInf}_{\mathcal{G}}^{\mathcal{H},\Lambda} \leq \frac{\Lambda \kappa C_{\mathrm{stab}}(m)}{n} + o(n^{-1}).$$

Combining with the first part proves

$$\Delta_{\mathcal{G}}^{\mathcal{H},\Lambda} \leq \frac{\Lambda \kappa C_{\mathrm{stab}}(m)}{n} + o(n^{-1}).$$

This proves Theorem 4.6. $\qquad\square$

# F. Proofs for Section 5

This section proves the attack-specific statements and the search–risk trade-off. The proofs use the decision-theoretic bounds of Appendix D and the RKME stability bounds of Appendix E.

## F.1. A useful weighted-normalization lemma

We first prove a lemma used by all power-posterior statements.

**Lemma F.1** (Stability of weighted normalization). *Let $\mathcal{T}$ be finite. Let $\rho$ and $\rho'$ be probability distributions on $\mathcal{T}$, and let $w : \mathcal{T} \to (0, \infty)$ satisfy $a \leq w(t) \leq b$ for all $t$. Define*

$$T_w \rho(t) = \frac{\rho(t) w(t)}{\sum_{s \in \mathcal{T}} \rho(s) w(s)}.$$

*Then*

$$\mathrm{TV}(T_w \rho, T_w \rho') \leq \frac{b}{a} \mathrm{TV}(\rho, \rho').$$

*Proof.* Let $\sigma = \rho - \rho'$ be the signed measure difference. Consider the path $\rho_\lambda = \rho' + \lambda \sigma$ for $\lambda \in [0, 1]$. Let $Z_\lambda = \sum_t \rho_\lambda(t) w(t)$. Since $a \leq w \leq b$, we have $Z_\lambda \geq a$.

For any test function $\varphi : \mathcal{T} \to [0, 1]$, define

$$F_\varphi(\lambda) = \sum_t \varphi(t) T_w \rho_\lambda(t).$$

Then

$$F_\varphi(\lambda) = \frac{\rho_\lambda(w\varphi)}{\rho_\lambda(w)}.$$

Differentiating,

$$F'_\varphi(\lambda) = \frac{\sigma(w\varphi) Z_\lambda - \rho_\lambda(w\varphi)\sigma(w)}{Z_\lambda^2}.$$

Let $c_\lambda = \rho_\lambda(w\varphi)/Z_\lambda$. Since $\varphi \in [0, 1]$, $c_\lambda \in [0, 1]$. Then

$$F'_\varphi(\lambda) = \frac{\sigma(w(\varphi - c_\lambda))}{Z_\lambda}.$$

The function $t \mapsto w(t)(\varphi(t) - c_\lambda)$ has oscillation at most $b$: its maximum is at most $b(1 - c_\lambda)$ and its minimum is at least $-b c_\lambda$. Therefore, by the dual characterization of total variation,

$$|\sigma(w(\varphi - c_\lambda))| \leq b \, \mathrm{TV}(\rho, \rho').$$

Since $Z_\lambda \geq a$,

$$|F'_\varphi(\lambda)| \leq \frac{b}{a} \mathrm{TV}(\rho, \rho').$$

Integrating over $\lambda \in [0, 1]$ gives

$$|F_\varphi(1) - F_\varphi(0)| \leq \frac{b}{a} \mathrm{TV}(\rho, \rho').$$

Taking the supremum over all $\varphi : \mathcal{T} \to [0, 1]$ proves the result. $\qquad\square \qquad\qquad\qquad \square$

## F.2. Proof of Theorem 5.1

In the loss-based membership game, the target is the source bit $T = B$ and the side information is the challenge sample $S^\star$. The model-only view $O_0 = (f, S^\star)$ already contains the model loss $\ell(f(X^\star), Y^\star)$ and any other information obtainable from the released model interface. The specification-assisted part of the attack is assumed to use RKME only through RKHS-compatible statistics of the form $\psi(\langle h_{S^\star}, \tilde{\mu}_D \rangle_{\mathcal{H}})$, with $\|h_{S^\star}\|_{\mathcal{H}} \leq \Lambda$ and $\psi$ Lipschitz.

Thus the attack belongs to the RKHS-compatible class controlled by Theorem 4.6. If the final randomized decision map has Lipschitz constant $C_{\text{loss}}$ with respect to the specification-dependent statistic, the bound in Theorem 4.6 is multiplied by $C_{\text{loss}}$. Hence

$$\Delta^{\mathcal{H},\Lambda}_{\text{loss-MIA}} \leq C_{\text{loss}} \frac{\Lambda \kappa C_{\text{stab}}(m)}{n} + o(n^{-1}).$$

This proves Theorem 5.1. $\qquad\square$

### F.3. Proof of Theorem 5.2

Fix the context $V$. For each $t \in \mathcal{T}$, define $w(t) = \exp\{-E_f(t; V)/\gamma\}$. Since $E_f(t; V) \in [0, B_E]$, we have $e^{-B/\gamma} \leq w(t) \leq 1$. The power posterior $q_R^\gamma(\cdot \mid V)$ is exactly the weighted normalization of $\rho_R(\cdot \mid V)$ by $w$, and $q_0^\gamma(\cdot \mid V)$ is the weighted normalization of $\rho_0(\cdot \mid V)$ by the same $w$. Applying Lemma F.1 with $a = e^{-B/\gamma}$ and $b = 1$ gives

$$\mathrm{TV}\left(q_R^\gamma(\cdot \mid V), q_0^\gamma(\cdot \mid V)\right) \leq e^{B_E/\gamma}\, \mathrm{TV}\left(\rho_R(\cdot \mid V), \rho_0(\cdot \mid V)\right).$$

This proves Eq. (27).

If the prior shift induced by $\rho_R$ is RKHS-compatible, then the same RKHS-smoothed TV and RKME stability argument as in Theorem 4.6 gives

$$\mathrm{TV}\left(\rho_R(\cdot \mid V), \rho_0(\cdot \mid V)\right) \leq C_{\text{post}} \frac{\Lambda \kappa C_{\text{stab}}(m)}{n} + o(n^{-1}),$$

where $C_{\text{post}}$ absorbs the Lipschitz constant of the prior map and finite-target constants. Multiplying by $e^{B_E/\gamma}$ yields

$$\Delta^{\mathcal{H},\Lambda}_{\text{post},\gamma} \leq C_{\text{post}} e^{B_E/\gamma} \frac{\Lambda \kappa C_{\text{stab}}(m)}{n} + o(n^{-1}).$$

This proves Theorem 5.2. $\qquad\square$

### F.4. Posterior membership inference

The posterior membership attack is the special case of Theorem 5.2 with $\mathcal{T} = \{0, 1\}$ and $T = B$. The context $V_{\text{mem}}$ contains the challenge sample, the model prediction, and the loss. The model energy $E_f(b; V_{\text{mem}})$ is a bounded score for the membership bit, and the RKME specification only changes the prior score $\rho_R(b \mid V_{\text{mem}})$. Therefore the same argument gives

$$\Delta^{\mathcal{H},\Lambda}_{\text{post-MIA},\gamma} \leq C_{\text{post}} e^{B_E/\gamma} \frac{\Lambda \kappa C_{\text{stab}}(m)}{n} + o(n^{-1}).$$

If the prior map is normalized so that its Lipschitz constant is one, then $C_{\text{post}} = 1$.

### F.5. Proof of Theorem 5.3

Let $s_R(a, \nu) = \langle h_{a,\nu}, \widetilde{\mu}_D \rangle_{\mathcal{H}}$ and $s_{R'}(a, \nu) = \langle h_{a,\nu}, \widetilde{\mu}_{D'} \rangle_{\mathcal{H}}$. If $\|h_{a,\nu}\|_{\mathcal{H}} \leq \Lambda$, then

$$|s_R(a, \nu) - s_{R'}(a, \nu)| \leq \Lambda \|\widetilde{\mu}_D - \widetilde{\mu}_{D'}\|_{\mathcal{H}}.$$

By Theorem 4.3, for neighboring datasets,

$$|s_R(a, \nu) - s_{R'}(a, \nu)| \leq \frac{2\Lambda \kappa C_{\text{stab}}(m)}{n} + o(n^{-1}).$$

Let $S_R(\nu) = \sum_{a \in \mathcal{A}} s_R(a, \nu)$ and $S_{R'}(\nu) = \sum_a s_{R'}(a, \nu)$. By assumption $S_R(\nu) \geq c_0$ and $S_{R'}(\nu) \geq c_0$. The induced priors are $\rho_R(a \mid \nu) = s_R(a, \nu)/S_R(\nu)$ and similarly for $\rho_{R'}$.

For nonnegative score vectors $s, s'$ with sums at least $c_0$,

$$\begin{aligned}
\|\rho_s - \rho_{s'}\|_1 &= \sum_a \left| \frac{s_a}{S} - \frac{s'_a}{S'} \right| \\
&\leq \sum_a \frac{|s_a - s'_a|}{S} + \sum_a s'_a \left| \frac{1}{S} - \frac{1}{S'} \right| \\
&\leq \frac{\|s - s'\|_1}{c_0} + \frac{|S - S'|}{c_0} \leq \frac{2\|s - s'\|_1}{c_0}.
\end{aligned}$$

Therefore

$$\mathrm{TV}(\rho_s, \rho_{s'}) = \frac{1}{2}\|\rho_s - \rho_{s'}\|_1 \leq \frac{\|s - s'\|_1}{c_0}.$$

Since $|\mathcal{A}| = K$ and each coordinate changes by at most $2\Lambda\kappa C_{\mathrm{stab}}(m)/n + o(n^{-1})$,

$$\mathrm{TV}\left(\rho_R(\cdot \mid \nu), \rho_{R'}(\cdot \mid \nu)\right) \leq \frac{2K\Lambda\kappa C_{\mathrm{stab}}(m)}{c_0 n} + o(n^{-1}).$$

Combining this prior-shift bound with the conditional-influence argument yields

$$\Delta_{\mathrm{AIA}}^{\mathcal{H},\Lambda} \leq \frac{2K\Lambda\kappa C_{\mathrm{stab}}(m)}{c_0 n} + o(n^{-1}).$$

If the prior is further passed through a power posterior with bounded energy in $[0, B_E]$, Theorem 5.2 multiplies the bound by $e^{B_E/\gamma}$. This proves Theorem 5.3. $\qquad\square$

## F.6. Proof of Lemma 5.4

By definition,

$$\|\widetilde{\mu}_D - \mu_P\|_{\mathcal{H}} \leq \|\widetilde{\mu}_D - \widehat{\mu}_D\|_{\mathcal{H}} + \|\widehat{\mu}_D - \mu_P\|_{\mathcal{H}} = \eta_m(D) + \|\widehat{\mu}_D - \mu_P\|_{\mathcal{H}}.$$

It remains to bound the empirical KME term. Let $\varphi(X) = k(X, \cdot)$. Then $\|\varphi(X)\|_{\mathcal{H}}^2 = k(X, X) \leq \kappa^2$. Also,

$$\widehat{\mu}_D - \mu_P = \frac{1}{n}\sum_{i=1}^n \left(\varphi(X_i) - \mu_P\right).$$

The expectation of its squared norm is

$$\mathbb{E}\|\widehat{\mu}_D - \mu_P\|_{\mathcal{H}}^2 = \frac{1}{n}\mathbb{E}\|\varphi(X) - \mu_P\|_{\mathcal{H}}^2$$

$$= \frac{1}{n}\left(\mathbb{E}\|\varphi(X)\|_{\mathcal{H}}^2 - \|\mu_P\|_{\mathcal{H}}^2\right) \leq \frac{\kappa^2}{n}.$$

Thus

$$\mathbb{E}\|\widehat{\mu}_D - \mu_P\|_{\mathcal{H}} \leq \frac{\kappa}{\sqrt{n}}.$$

Now consider the function

$$F(X_1, \ldots, X_n) = \|\widehat{\mu}_D - \mu_P\|_{\mathcal{H}}.$$

If one sample is replaced, $\widehat{\mu}_D$ changes by at most $2\kappa/n$ in RKHS norm. Hence $F$ has bounded differences $c_i = 2\kappa/n$. By McDiarmid's inequality,

$$\Pr[F - \mathbb{E}F \geq t] \leq \exp\left(-\frac{2t^2}{\sum_i c_i^2}\right) = \exp\left(-\frac{nt^2}{2\kappa^2}\right).$$

Taking $t = \kappa\sqrt{2\log(1/\delta)/n}$ gives, with probability at least $1 - \delta$,

$$\|\widehat{\mu}_D - \mu_P\|_{\mathcal{H}} \leq \frac{\kappa}{\sqrt{n}} + \kappa\sqrt{\frac{2\log(1/\delta)}{n}}.$$

Combining with the triangle inequality proves Lemma 5.4. $\qquad\square$

## F.7. Proof of Theorem 5.5

For each of the RKHS-compatible attacks in Section 5, the specification-side risk is bounded by a constant multiple of the RKME stability term. Specifically, Theorems 5.1, 5.2, and 5.3 all have the form

$$\Delta_{\mathcal{G}}^{\mathcal{H},\Lambda} \leq C_{\mathcal{G}}\frac{\Lambda\kappa C_{\mathrm{stab}}(m)}{n} + o(n^{-1}),$$

where $C_{\mathcal{G}}$ contains only attack-dependent constants such as the posterior temperature, normalization margin, target-space size, and Lipschitz constants of the prior or decision map.

On the search side, Lemma 5.4 gives

$$\|\widetilde{\mu}_D - \mu_P\|_{\mathcal{H}} \leq \eta_m(D) + \frac{\kappa}{\sqrt{n}} + \kappa\sqrt{\frac{2\log(1/\delta)}{n}}$$

with probability at least $1 - \delta$. Therefore $m$ affects the search side through the approximation term $\eta_m(D)$ and the risk side through the stability factor $C_{\text{stab}}(m)$. This proves Theorem 5.5. $\qquad\square$

### F.8. Gaussian RKME Regularity and the Square-root Size Bound

This subsection discusses the polynomial nondegeneracy condition used in Section 5 and proves Lemma 5.6 and Corollary 5.7. The purpose of polynomial nondegeneracy is not to assume the final risk bound. It is a primitive geometric regularity condition on the returned Gaussian RKME reduced set, from which the subquadratic growth of the projection-stability factor follows.

Recall that the local RKME stability factor is

$$C_{\text{stab}}(D, m) = \frac{\|D\Phi_m(\theta_D)\|_{\text{op}}^2}{\chi_D}, \qquad \chi_D = s_D^2 - \rho_D M_D.$$

Here $s_D^2 = \inf_{\|u\|=1} \|D\Phi_m(\theta_D)[u]\|_{\mathcal{H}}^2$, $\rho_D = \|r_D\|_{\mathcal{H}}$, and $M_D = \sup_{\|u\|=1} \|D^2\Phi_m(\theta_D)[u, u]\|_{\mathcal{H}}$. Thus, to obtain subquadratic stability, it suffices to control the growth of the numerator $\|D\Phi_m(\theta_D)\|_{\text{op}}^2$ and prevent the curvature-residual margin $\chi_D$ from degenerating too fast.

For Gaussian RKME, the numerator is controlled by the smoothness of the Gaussian feature map. Let $k_\sigma(x, z) = \exp(-\|x - z\|^2/(2\sigma^2))$ and write $\phi(z) = k_\sigma(z, \cdot)$. The Gaussian feature map has uniformly bounded first- and second-order RKHS derivatives. Consequently, when the RKME weights are uniformly bounded, the squared tangent-operator norm grows at most linearly in the number of reduced atoms. The remaining issue is whether the reduced-set tangent directions become nearly singular as $m$ grows. Polynomial nondegeneracy rules out such rapid degeneration.

Let

$$\Gamma_D = D\Phi_m(\theta_D)^* D\Phi_m(\theta_D)$$

be the tangent Gram matrix at the returned RKME solution. Under the Euclidean norm on the local reduced-set parameter space, $s_D^2 = \lambda_{\min}(\Gamma_D)$. We say that the returned Gaussian RKME solutions are polynomially nondegenerate up to size $m_*$ if there exist constants $c_s > 0$, $\tau \in (0, 1)$, $a < 1$, and $\beta_{\max} < \infty$ such that, for all relevant datasets and all $1 \leq m \leq m_*$,

$$|\beta_j| \leq \beta_{\max}, \qquad \lambda_{\min}(\Gamma_D) \geq c_s m^{-a}, \qquad \rho_D M_D \leq (1 - \tau)\lambda_{\min}(\Gamma_D).$$

The first condition prevents unbounded weights. The second condition excludes nearly singular tangent directions, which may arise from duplicate or highly redundant reduced atoms. The third condition ensures that the residual-curvature term in the Hessian does not overwhelm the positive tangent curvature. Together they imply $\chi_D \geq \tau\lambda_{\min}(\Gamma_D)$.

This condition is natural for Gaussian RKME because degeneracy of the tangent Gram matrix corresponds to an explicit geometric failure. At any fixed size $m$, if the reduced atoms are distinct and the effective location directions are not removed by zero weights, the Gaussian tangent functions are linearly independent. Indeed, suppose that

$$\sum_{j=1}^{m} a_j k_\sigma(z_j, \cdot) + \sum_{j=1}^{m}\sum_{\ell=1}^{d} b_{j\ell}\, \partial_{z_\ell} k_\sigma(z_j, \cdot) = 0$$

as an RKHS function. Taking the Fourier transform in the second argument gives, up to a nonzero Gaussian factor,

$$\sum_{j=1}^{m} \left(a_j - ib_j^\top \omega\right) e^{-i\omega^\top z_j} = 0, \qquad \forall \omega \in \mathbb{R}^d,$$

where $b_j = (b_{j1}, \ldots, b_{jd})$. Choose a generic direction $v$ such that $v^\top z_1, \ldots, v^\top z_m$ are distinct, and set $\omega = tv$. Then

$$\sum_{j=1}^{m} \left( a_j - itb_j^\top v \right) e^{-itv^\top z_j} = 0, \qquad \forall t \in \mathbb{R}.$$

Exponential polynomials with distinct frequencies are linearly independent; hence $a_j = 0$ and $b_j^\top v = 0$ for all $j$. Repeating the argument for $d$ linearly independent generic directions yields $b_j = 0$ for all $j$. Therefore, exact tangent singularity can occur only through degenerate configurations, such as coincident atoms or ineffective location directions. Polynomial nondegeneracy is the quantitative requirement that the returned RKME solutions do not approach these degenerate configurations faster than a polynomial rate.

We now prove Lemma 5.6.

**Proof of Lemma 5.6.** For the Gaussian feature map $\phi(z) = k_\sigma(z, \cdot)$, the reproducing identity gives

$$\langle \phi(z), \phi(z') \rangle_{\mathcal{H}} = k_\sigma(z, z').$$

For any $v \in \mathbb{R}^d$,

$$\|D\phi(z)[v]\|_{\mathcal{H}}^2 = \partial_t \partial_s k_\sigma(z + tv, z + sv)|_{t=s=0}.$$

Since

$$k_\sigma(z + tv, z + sv) = \exp\left( -\frac{(t-s)^2 \|v\|^2}{2\sigma^2} \right),$$

we obtain

$$\|D\phi(z)[v]\|_{\mathcal{H}} = \frac{\|v\|}{\sigma}.$$

Similarly,

$$\|D^2\phi(z)[v, v]\|_{\mathcal{H}}^2 = \partial_t^2 \partial_s^2 k_\sigma(z + tv, z + sv)\big|_{t=s=0} = \frac{3\|v\|^4}{\sigma^4}.$$

Thus

$$\|D^2\phi(z)[v, v]\|_{\mathcal{H}} = \frac{\sqrt{3}}{\sigma^2} \|v\|^2.$$

Let $u = (a_1, \ldots, a_m, v_1, \ldots, v_m)$ be a parameter direction satisfying

$$\sum_{j=1}^{m} a_j^2 + \sum_{j=1}^{m} \|v_j\|^2 = 1.$$

The first derivative of the reduced-set map is

$$D\Phi_m(\theta_D)[u] = \sum_{j=1}^{m} a_j \phi(z_j) + \sum_{j=1}^{m} \beta_j D\phi(z_j)[v_j].$$

Hence

$$\|D\Phi_m(\theta_D)\|_{\mathrm{op}}^2 = \lambda_{\max}(\Gamma_D) \le \mathrm{tr}(\Gamma_D).$$

The $m$ weight columns have RKHS norm $\|\phi(z_j)\|_{\mathcal{H}} = 1$. For each atom, the $d$ location-derivative columns have squared RKHS norm at most $\beta_{\max}^2/\sigma^2$. Therefore

$$\mathrm{tr}(\Gamma_D) \le m + md\frac{\beta_{\max}^2}{\sigma^2} = C_\Phi m, \qquad C_\Phi = 1 + \frac{d\beta_{\max}^2}{\sigma^2}.$$

Thus

$$\|D\Phi_m(\theta_D)\|_{\mathrm{op}}^2 \le C_\Phi m.$$

We also record that the Gaussian second-derivative bound implies a uniform bound on $M_D$. Indeed,

$$D^2\Phi_m(\theta_D)[u, u] = 2\sum_{j=1}^m a_j D\phi(z_j)[v_j] + \sum_{j=1}^m \beta_j D^2\phi(z_j)[v_j, v_j].$$

Using Cauchy's inequality and the derivative bounds above,

$$\|D^2\Phi_m(\theta_D)[u, u]\|_{\mathcal{H}} \le \frac{2}{\sigma}\sum_{j=1}^m |a_j|\|v_j\| + \frac{\sqrt{3}\beta_{\max}}{\sigma^2}\sum_{j=1}^m \|v_j\|^2$$

$$\le \frac{1}{\sigma}\left(\sum_{j=1}^m a_j^2 + \sum_{j=1}^m \|v_j\|^2\right) + \frac{\sqrt{3}\beta_{\max}}{\sigma^2}\sum_{j=1}^m \|v_j\|^2$$

$$\le \frac{1}{\sigma} + \frac{\sqrt{3}\beta_{\max}}{\sigma^2}.$$

Thus $M_D$ is uniformly bounded for bounded weights. The polynomial nondegeneracy condition further gives

$$s_D^2 = \lambda_{\min}(\Gamma_D) \ge c_s m^{-a}$$

and

$$\rho_D M_D \le (1 - \tau)\lambda_{\min}(\Gamma_D) = (1 - \tau)s_D^2.$$

Therefore

$$\chi_D = s_D^2 - \rho_D M_D \ge \tau s_D^2 \ge \tau c_s m^{-a}.$$

Combining the numerator and denominator bounds yields

$$C_{\text{stab}}(D, m) = \frac{\|D\Phi_m(\theta_D)\|_{\text{op}}^2}{\chi_D} \le \frac{C_\Phi m}{\tau c_s m^{-a}} = \frac{C_\Phi}{\tau c_s}m^{1+a}.$$

Let $C_0 = C_\Phi/(\tau c_s)$ and set $\xi = 1 - a > 0$. Then $1 + a = 2 - \xi$, so

$$C_{\text{stab}}(D, m) \le C_0 m^{2-\xi}.$$

Taking the uniform upper bound over the relevant datasets gives the stated bound for $C_{\text{stab}}(m)$. This proves Lemma 5.6.

**Proof of Corollary 5.7.** By Theorem 5.5, for the RKHS-compatible learnware-inversion attacks considered in Section 5,

$$\Delta_{\mathcal{G}}^{\mathcal{H},\Lambda}(m) \le C_{\mathcal{G}}\frac{\Lambda\kappa C_{\text{stab}}(m)}{n} + o(n^{-1}).$$

Set $m_* = \lfloor\sqrt{n}\rfloor$. By Lemma 5.6, there exist $C_0 > 0$ and $\xi > 0$ such that

$$C_{\text{stab}}(m_*) \le C_0 m_*^{2-\xi}.$$

Since $m_*^2 \le n$, we have

$$\frac{C_{\text{stab}}(m_*)}{n} \le \frac{C_0 m_*^{2-\xi}}{n} \le C_0 m_*^{-\xi}.$$

Substituting this into the risk decomposition gives

$$\Delta_{\mathcal{G}}^{\mathcal{H},\Lambda}(m_*) \le C_{\mathcal{G}}\Lambda\kappa C_0 m_*^{-\xi} + o(n^{-1}).$$

For the utility side, Lemma 5.4 gives

$$\|\widetilde{\mu}_D - \mu_P\|_{\mathcal{H}} \le \eta_{m_*}(D) + O_p(n^{-1/2}).$$

Since $m_* = \lfloor \sqrt{n} \rfloor$, we have $n^{-1/2} = O(m_*^{-1})$. If $\eta_m(D) = O(m^{-\alpha})$, then $\eta_{m_*}(D) = O(m_*^{-\alpha})$. Therefore

$$\|\widetilde{\mu}_D - \mu_P\|_{\mathcal{H}} = O(m_*^{-\alpha}) + O_p(m_*^{-1}).$$

This proves Corollary 5.7.

The risk-tolerance statement in the main text follows immediately. For any $\varepsilon > 0$, if

$$m_* \geq \left( \frac{C_{\mathcal{G}} \Lambda \kappa C_0}{\varepsilon} \right)^{1/\xi},$$

then $C_{\mathcal{G}} \Lambda \kappa C_0 m_*^{-\xi} \leq \varepsilon$, and hence

$$\Delta_{\mathcal{G}}^{\mathcal{H}, \Lambda}(m_*) \leq \varepsilon + o(n^{-1}).$$

