# OpenReview forum: "A Statistical Framework for Analyzing Specification Resistance to Learnware-Inversion Risks"
_ICML.cc/2026/Conference — ICML 2026 regular_

### Official Review · Reviewer_3en1 · 2026-03-10

**Soundness:** 2
**Presentation:** 3
**Significance:** 1
**Originality:** 3
**Overall Recommendation:** 4
**Confidence:** 2

**Summary:**

The paper investigates the privacy risks of releasing machine learning models accompanied by their specifications in the learnware paradigm. To quantify this learnware-inversion risk, the authors propose a game-theoretic framework utilizing variational inference and geometry analysis. Theoretical analyses are conducted to demonstrate that an appropriately sized RKME specification introduces negligible extra privacy risk.

**Compliance With Llm Reviewing Policy:**

Affirmed.

**Final Justification:**

Regarding the Gaussian assumption in Theorem 3.4, the authors clarified that it is intended as an analytically convenient special case and that the broader posterior-based framework could, in principle, extend to richer score models. However, this does not fully address my original concern about realism in practice, since the rebuttal does not discuss whether commonly used MIA scores, such as loss or entropy-based scores, actually satisfy any reasonable and analyzable distributional family in this setting, nor does it provide empirical evidence supporting such a modeling choice. On the worst-case issue, the second-round rebuttal provided a more refined explanation by relating the distribution-level notion to local risk and arguing that small average risk limits the prevalence of highly vulnerable points. I appreciate this clarification, but it still does not fully resolve the concern that vulnerable individual points may be obscured by averaging. Overall, the rebuttal clarified the scope but did not substantially change my view of the paper’s main limitations. In addition, the learnware setting is relatively specialized, so while the problem is technically interesting, I find the broader significance of analyzing privacy in this setting somewhat limited. For these reasons, I will maintain my current score.

**Key Questions For Authors:**

The paper models the RKME specification as $\hat{\mathcal{P}}(x)$, which approximates the marginal distribution $p(x)$. However, many membership inference attacks such as LiRA and RMIA [2,3] assume that the attacker may already have access to the data distribution. In this case, the attacker may already know $p(x)$. Could the authors clarify what additional information the RKME specification provides beyond this assumption?


[2] Membership inference attacks from first principles. IEEE S&P, 2022.

[3] Low-cost high-power membership inference attacks. ICML, 2024.

**Strengths And Weaknesses:**

Pros
1. The motivation is novel. It is the first formal study to analyze the joint privacy risk of releasing both a model and its specification, rather than isolating the specification.
2. The theoretical foundation is solid. The authors conduct rigorous mathematical proofs, successfully decoupling the incremental specification risk from the inherent model risk using power posteriors.

Cons
1. The theoretical assumptions are overly strong. For example, Theorem 3.4 assumes that prediction errors follow a Gaussian distribution with mean $0$, which may be unrealistic for commonly used MIA scores.
2. The paper lacks discussion on worst-case privacy risks [1]. The proposed metric $Risk_f(R_f)$ bounds the expected risk over the global data distribution (e.g., relying on expectations such as $\mathbb{E}_{s \sim \mathcal{P}}$ in Theorem 3.2), which may obscure the privacy risks faced by the most vulnerable data points.
3. The paper does not provide concrete attack methods that explicitly leverage the specification. It would be helpful to include specific attack strategies and corresponding experiments to better illustrate how the specification could increase privacy risk.

[1] Evaluations of machine learning privacy defenses are misleading. CCS, 2024.

---

> ### Author Rebuttal · Authors · 2026-03-30
>
> Many thanks for the constructive reviews!
>
> ---
>
> **Q1**: ... Theorem 3.4 assumes that prediction errors follow a Gaussian distribution with mean 0 ...
>
> **A1**: Thanks for the feedback! We would like to clarify that the Gaussian assumption is used only in Theorem 3.4. Our other main results do not rely on any additional distributional assumption.
>
> For Theorem 3.4, the Gaussian assumption is mainly introduced to make the posterior-inference analysis tractable and to obtain an explicit closed-form characterization of the specification risk. This type of assumption is also common in privacy analysis when one aims to derive analytically interpretable bounds from error-based attacks. More importantly, handling fully general error distributions in posterior-based MIA remains challenging in all the theoretical analysis of such attacks. However, the general analysis pipeline remains applicable: one can replace the Gaussian likelihood with a more flexible estimate of the error distribution, such as a nonparametric density estimator or a richer parametric family (e.g., mixture models), and then perform the same posterior-based inference procedure.
>
> Therefore, we view Theorem 3.4 as a clean first-step theoretical result for a standard case, while the extension to more general error distributions is an important direction for future work. We will clarify this scope and limitation more explicitly in the revision.
>
> ---
>
> **Q2**: The paper lacks discussion on worst-case privacy risks.
>
> **A2**: Thank you for this important comment. Current distribution-level risk is intentional, and is closely tied to the learnware paradigm itself rather than being a fallback due to technical difficulty.
>
> In the learnware setting, a specification is fundamentally a distribution-level summary of the developer’s training data, and its primary role is to support distribution-aware search, identification, and reuse in the learnware market. Accordingly, the privacy question studied in this paper is also distribution-level: namely, how much additional adversarial advantage is introduced by releasing such a specification together with the model under the underlying data-generating distribution $P$. In this sense, our choice of averaging over $s \sim P$ is not incidental, but is aligned with the way learnwares are represented, searched, and deployed.
>
> By contrast, a pointwise worst-case notion asks for the risk of the single most vulnerable sample. This is certainly an important privacy perspective, but it addresses a different objective. For this reason, we believe that the present distribution-level formulation is the natural first step for formalizing specification risk in learnware, while worst-case or local privacy notions are valuable complementary directions for future work.
>
> ---
>
> **Q3**:  It would be helpful to include corresponding experiments ...
>
> **A3**: Thanks for the feedback! We have added validation experiments. Due to the rebuttal word limit, detailed information about these supplementary experiments and results can be found in our response (**A1**) to Reviewer PhxA.
>
> ---
>
> **Q4**: ... Could the authors clarify what additional information the RKME specification provides beyond this assumption?
>
> **A4**: Thank you for this important question. We agree that the RKME specification in our framework plays exactly the role of a prior on the marginal distribution $p(x)$. Therefore, if an attacker already has access to the true marginal distribution, then the additional information provided by the specification should indeed become much smaller. This is fully consistent with our formulation, rather than a contradiction to it.
>
> More precisely, our notion of specification risk is defined as the incremental adversarial advantage introduced by releasing the specification beyond the model alone. In our game-based framework, any such prior knowledge available to the attacker can naturally be incorporated into the side information $\nu$. If $\nu$ already contains the true $p(x)$, then the marginal contribution of the RKME specification is expected to decrease accordingly, possibly becoming negligible.
>
> The point of our analysis is therefore not to assume that the attacker has no prior knowledge at all, but to quantify how much extra information is introduced by the released specification in the learnware setting. In practice, the publicly released RKME is precisely the mechanism through which a developer-specific approximation to the training-data marginal becomes available to the attacker. Hence, the specification remains meaningful exactly in the regime where the attacker does not already know the true marginal distribution, or only has an imperfect approximation to it.
>
> ---
>
> We sincerely thank the reviewer again for the careful and thoughtful review! If our responses have helped clarify the paper and address the reviewer’s concerns, we would respectfully appreciate a reconsideration of the current confidence level and score!

---

> > ### Author Rebuttal · Reviewer_3en1 · 2026-04-01
> >
> > Thank you for the reply. The responses helped clarify the paper’s intended scope, especially regarding the role of RKME as side information and the distribution-level nature of the proposed risk. However, my main concerns are only partially addressed, not fully resolved.
> >
> > In particular, the response to Theorem 3.4 clarifies why the Gaussian (with mean 0) assumption is adopted, but does not address my concern about its realism for practical MIA scores. The discussion on worst-case privacy risk explains why the paper focuses on distribution-level risk, but does not address whether vulnerable individual points may still be obscured by this averaging.
> >
> > Overall, the rebuttal improves clarity, but it does not sufficiently change my assessment of the paper’s current limitations. I therefore will maintain my current score.

---

> > > ### Author Response · Authors · 2026-04-01
> > >
> > > Dear Reviewer 3en1,
> > >
> > > We sincerely thank you for taking the time to read our rebuttal so carefully, and we are very glad that it has addressed most of your concerns!
> > >
> > > That said, we understand that you still have further concerns on two points, and we are very happy to provide additional clarification below!
> > >
> > > ---
> > >
> > > **A1**: The framework extends beyond Gaussian score models. Let
> > > $q_0(\epsilon)$ and $q_1(\epsilon)$
> > > denote the conditional densities of the attack score under $\tau=0$ and $\tau=1$, respectively. Then the Bayes-optimal posterior attack is always
> > > $\hat\tau(\epsilon)=\arg\max_{b\in\{0,1\}} \pi_b q_b(\epsilon)$,
> > > where $\pi_b$ is the prior of class $b$. Under equal priors, its success probability is
> > > $\mathrm{Gain}^*=\int \max\{q_0(\epsilon),q_1(\epsilon)\}\,d\epsilon
> > > = \frac{1}{2}+\frac{1}{2}\mathrm{TV}(q_0,q_1)$,
> > > where $\mathrm{TV}(q_0,q_1)=\frac{1}{2}\|q_0-q_1\|_1$ is the total variation distance. Therefore, the posterior-based MIA analysis does not fundamentally rely on Gaussianity: what matters is how distinguishable the two score distributions are. The Gaussian assumption is simply the special case in which this distinguishability admits a particularly clean closed-form expression.
> > >
> > > This observation immediately covers several richer score families. For non-centered Gaussians, as noted above, the same likelihood-ratio rule applies with a shifted threshold. For Laplace-type score models,
> > > $q_b(\epsilon)=\frac{1}{2\lambda_b}\exp(-|\epsilon-\mu_b|/\lambda_b)$,
> > > the posterior decision is again explicit, now with a piecewise-linear log-likelihood ratio. For finite Gaussian mixtures,
> > > $q_b(\epsilon)=\sum_{j=1}^{m_b} w_{b,j}\phi(\epsilon;\mu_{b,j},\sigma_{b,j}^2)$,
> > > the Bayes rule compares two mixture likelihoods; although the resulting threshold is no longer a single closed-form scalar, the same posterior-based analysis still applies. More generally, for arbitrary score families one can work directly with divergence quantities such as total variation, Hellinger distance, or KL divergence. For example,
> > > $\mathrm{TV}(q_0,q_1)\le \sqrt{\frac{1}{2}\mathrm{KL}(q_0\|q_1)}$,
> > > so one may upper bound the attack success even without a Gaussian closed form.
> > >
> > > In this sense, Theorem 3.4 is best understood as the first explicit closed-form theorem for a posterior-based MIA under a simple score model. The framework itself is more general: replacing the Gaussian error model by other practical score distributions changes the final formula, but not the underlying mechanism. We will clarify this point in the revision so that Theorem 3.4 is interpreted as a clean analytically tractable instance, rather than as an exact realism claim about all practical MIA scores.
> > >
> > >
> > >
> > > **A2**: We would like to emphasize that the present formulation is still informative about worst-case behavior in a nontrivial way.
> > >
> > > To make this precise, let us define the local incremental risk at a point $s$ by
> > > $r(s)=\sup_A \Pr(A(f,R_f,\nu(s))=\tau(s)\mid s)-\sup_{A'} \Pr(A'(f,\nu(s))=\tau(s)\mid s)$.
> > > Then the distribution-level quantity studied in our paper is exactly the expectation of this pointwise quantity under the data-generating distribution:
> > > $\mathrm{Risk}\_f(R\_f)=\mathbb{E}\_{s\sim P}[r(s)]$
> > > (up to the same adversarial optimization as in our definition). Therefore, our notion does not “erase” vulnerable points; rather, it averages their local contribution. This already implies a useful prevalence bound: for any threshold $\lambda>0$, if
> > > $V_\lambda=\{s:r(s)\ge \lambda\}$,
> > > then by Markov’s inequality,
> > > $P(V_\lambda)\le \frac{\mathbb{E}[r(s)]}{\lambda}=\frac{\mathrm{Risk}_f(R_f)}{\lambda}$.
> > > Hence, a small global risk guarantees that highly vulnerable points cannot occupy a large fraction of the population. In other words, averaging does not rule out all bad points, but it does control how prevalent they can be.
> > >
> > > A similar observation holds at the subgroup level. For any measurable subset $A\subseteq Z$, define the conditional risk
> > > $\mathrm{Risk}_A=\mathbb{E}[r(s)\mid s\in A]$.
> > > Then
> > > $\mathrm{Risk}\_f(R\_f)=P(A)\mathrm{Risk}\_A+P(A^c)\mathrm{Risk}\_{A^c}$.
> > > Consequently,
> > > $\mathrm{Risk}_A\le \frac{\mathrm{Risk}_f(R_f)}{P(A)}$.
> > > Thus, if a subgroup $A$ has non-negligible probability mass, its conditional risk cannot be arbitrarily large while the overall distribution-level risk remains small. This shows that our theory already gives a partial worst-case message: although it does not certify every single point uniformly, it does prevent large high-risk regions or large high-risk subpopulations from being hidden by averaging.
> > >
> > > ---
> > >
> > > We sincerely thank you again for your continued attention to our paper. We remain very happy and fully prepared to respond to any further questions you may have! If our additional clarifications fully address your concerns, we would respectfully invite you to reconsider the current score or confidence level, which would mean a great deal to us!

---

### Official Review · Reviewer_vxmg · 2026-03-11

**Soundness:** 2
**Presentation:** 2
**Significance:** 2
**Originality:** 3
**Overall Recommendation:** 4
**Confidence:** 4

**Summary:**

This paper presents the first formal study of the risks that arise when a specification is attached to a model, as opposed to releasing the model alone.

**Compliance With Llm Reviewing Policy:**

Affirmed.

**Final Justification:**

The rebuttal addresses most of my concerns and explains the question I asked. Although the attribute inference is white-box, a black-box setting can also be applied. I changed my score to weak accept.

**Key Questions For Authors:**

Why can DP not be directly applicable in this paper?

**Limitations:**

Yes

**Strengths And Weaknesses:**

## Strength:

### 1. A unified theoretical framework.

The paper proposes a unified theoretical framework to analyze the additional privacy risk caused by specifications.

### 2. Insightful connection between specification size and privacy risk.

The paper studies the relationship between the size of the RKME specification and the potential privacy risk.


## Weakness:

### 1. Basis expansion assumption

In Lemma D.2, the proof assumes that the empirical KME can be written as an infinite sum of kernel functions from a countable set. However, separability of a Hilbert space only means that a countable dense set exists. It does not mean that these kernel functions form a basis that can represent every element as a series. Therefore, the step
$\frac{1}{n}\sum_{i=1}^n k(x_i,\cdot)=\sum_{i=1}^{\infty}\alpha_i k(s_i,\cdot)$
is not clearly justified. Since the uniqueness argument depends on this expansion, the proof appears incomplete.

### 2. Truncation vs best approximation

In Lemma D.2 Step 1, the proof treats the truncated series
$\sum_{i=1}^{m-1}\alpha_i k(s_i,\cdot)$
as the best approximation of the empirical KME in an (m-1)-dimensional space. However, this is only true when the functions form an orthogonal basis. For a general set of kernel functions, truncating coefficients does not guarantee the smallest error.

### 3. Volume vs probability

In the proof of Theorem 4.2, the authors compare the volume on the dataset manifold with the total manifold volume and interpret this ratio as a probability. However, datasets are generated from the distribution $P^n$, not from a uniform distribution on the manifold. Therefore, the probability of the bad event should depend on the density $p(D)$. A small geometric volume does not always mean a small probability. This step needs further justification.


### 4. manifold dimension $n-2m$

The paper claims that the set of datasets producing the same RKME forms a manifold with dimension $n-2m$. This result depends on the Jacobian of a mapping having full row rank almost everywhere. However, the Jacobian entries come from Gaussian kernel functions and their derivatives, which may be strongly related to each other. Because of this structure, the rank condition may fail in many cases. The proof should explain more clearly why the full-rank assumption holds.

### 5. Experiments.

I didn't see any experiments in this paper. This is very important to demonstrate that your proof is correct.

### 6. Black-box

This paper claims they only use black-box oracle query. However, attribute inference is a white-box attack.

---

> ### Author Rebuttal · Authors · 2026-03-30
>
> Many thanks for the valuable reviews!
>
> ---
>
> **Q1 & Q2**: Basis expansion assumption & Truncation vs best approximation
>
> **A1 & A2**: Thank you for this careful comment. We understand that both concerns are related to whether the basis construction used in the Gaussian RKHS is sufficiently justified.
>
> A more formal way to present this step is the following. Since the Gaussian RKHS is a separable Hilbert space, one may start from a countable dense linearly independent family of kernel sections, for example ${k(q,\cdot): q\in\mathbb{Q}}$, and apply Gram--Schmidt to obtain a countable orthonormal basis ${e_i}$. Any $f$ in the RKHS then admits the standard expansion
>  $f=\sum_{i=1}^{\infty}\langle f,e_i\rangle e_i$.
>
> For Lemma D.2, this orthonormal-basis formulation is already sufficient for the projection argument we use, including the decomposition of the empirical KME and the construction of the relevant finite-dimensional subspaces. In other words, the essential point is the existence of a countable Hilbert basis and the associated projection structure, rather than treating the original kernel sections themselves as the primary expansion basis. The current appendix presents this step in a compressed form, which may make basis construction appear less explicit than intended. We thank the reviewer for pointing this out.
>
> ---
>
> **Q3**: Volume vs probability
>
> **A3**: Thanks for the feedback. However, this issue does not arise in our setting, since we do not assume a uniform distribution on the manifold. Rather, we consider an arbitrary continuous distribution. When $P$ admits a density $p$, the product measure $P^n$ is absolutely continuous with respect to the ambient Lebesgue measure. Hence, any set of Lebesgue measure zero, i.e., a geometrically negligible set in the dataset space, also has probability zero under $P^n$. This is exactly the justification used in our argument.
>
> ---
>
> **Q4**: manifold dimension $n-2m$
>
> **A4**: Thanks for the feedback. We understand the reviewer’s concern. However, in our setting, this issue does not arise because the full-row-rank condition is not assumed but established in the appendix.
>
> The full-row-rank condition is not assumed in our paper, it is established in the appendix. More precisely, Corollary D.9 reduces the manifold statement to showing that the Jacobian is of full row rank almost everywhere, and Proposition D.10 is devoted exactly to this point. The point of Proposition D.10 is precisely that such degenerate configurations form a measure-zero set, and hence the Jacobian is of full row rank almost everywhere. Therefore, the manifold dimension claim in Proposition 4.1 is based on an a.e. rank statement proved in the appendix, rather than on an unverified assumption.
>
> ---
>
> **Q5 & Q6**: Experiments & Black-box
>
> **A5 & A6**: We have added supplementary validation experiments and provided a more detailed explanation of the motivation and necessity of adopting the black-box setting. Due to the rebuttal word limit, detailed information about these additional experiments and their results can be found in our response to Reviewer PhxA (**A1** & **A2**).
>
> ---
>
> **Q7**: Why can DP not be directly applicable in this paper?
>
> **A7**: Our theory and DP are related in that they are both analytical frameworks, but they are formulated for different objects of study. In DP, privacy protection arises from algorithmic randomness: adding noise makes it hard for an adversary to distinguish individuals in the original data. In contrast, the privacy protection of RKME in our setting comes from data compression. Because the RKME generation algorithm is deterministic, DP’s analysis framework does not apply directly. Moreover, the RKME size is typically small; injecting noise into the RKME generation would substantially worsen the utility–privacy trade‑off familiar in DP and would degrade Learnware retrieval. Thus, DP is not the right tool for our object of study here. However, our theory shows that even without added noise, a compression mechanism like RKME can protect the privacy of the original data. For this reason, although the goals of DP and our work are **aligned**, the objects they analyze are fundamentally different.
>
> On the other hand, our contribution is that we are the first to provide a theoretical, quantitative analysis of how an auxiliary dataset, together with a model, exposes private information. Existing work on model inversion attacks that incorporate auxiliary datasets mainly focuses on developing more effective attack algorithms, while the theoretical modeling of why such auxiliary datasets make models more prone to privacy leakage, and how to quantify this, has been entirely missing. Our results are novel in the theory of model inversion with auxiliary data.
>
> ---
>
> We sincerely thank the reviewer again for the careful review. If our responses have addressed the concerns and improved the reviewer’s assessment of the paper, we respectfully ask the reviewer to raise the score.

---

> > ### Author Rebuttal · Reviewer_vxmg · 2026-04-01
> >
> > I think most of my concerns have been addressed. I will increase the score to weak accept.

---

> > > ### Author Response · Authors · 2026-04-01
> > >
> > > Dear Reviewer vxmg,
> > >
> > > Thank you so much for your kind reply and for adjusting the score! We are very grateful for your constructive comments, which have helped us think more carefully about the technical details of the paper.
> > >
> > > We will further improve and polish the paper in future versions based on your suggestions. We also very much appreciate the opportunity for further discussion, and we remain happy to respond to any additional questions or concerns!
> > >
> > > Best
> > >
> > > Authors

---

### Official Review · Reviewer_PhxA · 2026-03-13

**Soundness:** 2
**Presentation:** 3
**Significance:** 3
**Originality:** 3
**Overall Recommendation:** 4
**Confidence:** 2

**Summary:**

This paper studies the privacy risks that arise when a specification is released together with a model in the learnware paradigm. The authors propose a game-theoretic framework to formalize learnware-inversion attacks and analyze the additional risk introduced by specifications. By combining variational inference and geometric analysis, the paper derives quantitative bounds on the specification risk and provides theoretical guarantees for the data protection ability of the commonly used RKME specification.

**Compliance With Llm Reviewing Policy:**

Affirmed.

**Final Justification:**

The authors’ rebuttal and additional experiments have addressed my major concerns. Accordingly, I raise my rating to Weak Accept.

**Key Questions For Authors:**

Beyond membership inference and attribute inference attacks, what other types of attacks could pose risks in the learnware setting?
For example, could attacks such as model inversion attacks exploit both the model and the specification to recover information about the original training data? It would be helpful if the authors could discuss whether the proposed framework or theoretical results extend to such attacks.

**Limitations:**

Yes

**Strengths And Weaknesses:**

**Strengths:**

1. The idea is novel and timely. This is the first work that analyzes the privacy risks in the learnware paradigm when an adversary can access both the specification and the model. This problem is important because specifications are designed to summarize training data distributions while enabling model reuse.

2. The paper is well organized and the presentation is clear. The logical flow from the learnware framework, to the definition of learnware-inversion attacks, and then to the theoretical analysis is easy to follow.

3. The theoretical framework and derivations appear technically solid. The paper provides a principled formulation of specification risk and derives bounds under two attack scenarios.

**Weaknesses:**

1. The paper does not include any empirical experiments to validate the theoretical findings. The authors could evaluate the proposed claims using existing learnware systems or even a small simulated setup to test whether common attacks such as membership inference or attribute inference indeed fail when RKME specifications are used. The absence of such empirical evidence raises concerns about the completeness and practical relevance of the work.

2. As stated in the paper, the analysis assumes a black-box adversary that can only interact with the model through queries. However, this assumption may not fully reflect realistic learnware scenarios. According to the definition of learnware, users are expected to reuse the released model for downstream tasks, which typically implies white-box access to model parameters. This introduces additional attack surfaces, including gradient-based or parameter-based attacks. As a result, the practical insights provided by the current theoretical analysis may be limited.

---

> ### Author Rebuttal · Authors · 2026-03-30
>
> Many thanks for the valuable reviews!
>
> ---
>
> **Q1**: The paper does not include empirical experiments.
>
> **A1**: Thanks for your feedback! We would like to respectfully clarify to reviewers that general privacy-theory work like ours is often not well-suited to empirical validation, because the results are model-agnostic and distribution-agnostic. Verifying such results on a particular dataset and model does not increase the credibility of the theory, and we believe the theoretical contribution of this paper is sufficient and complete.
>
> We have conducted following validation experiments.
>
> **Datasets**
>
> We use six real-world datasets: Postures [Gardner et al. 2014], Bank [Moro, Cortez, and Rita 2014], Mushroom [Wagner, Heider, and Hattab 2021], PPG-DaLiA [Reiss et al. 2019], PFS [Kaggle 2018], and M5 [Makridakis, Spiliotis, and Assimakopoulos 2022]. These datasets span various tasks and scenarios, varying in scale from **550 thousand to 46 million** instances. We have developed a learnware market prototype comprising about **4000 models** of various types.
>
> **Adversary**
>
> We evaluate the risk induced under three adversaries. The first two correspond to Adversary 3.1 and Adversary 3.7 in Section 3. To demonstrate the generality of our theory, we also consider a third, widely used auxiliary-data attack: the shadow-model MIA of Shokri et al. (2017).
>
> **Valuation**
>
> For search ability, we employ error rate and root-mean-square error (RMSE) as the loss function for classification and regression scenarios, respectively, collectively referred to as Search error. A **smaller search error** indicates **stronger search ability**. For privacy risk introduced in paper, a **smaller privacy risk** indicates **stronger privacy protection**.
>
> Due to space limitations, our results are provided in the anonymized link below. https://default-anno-bucket.s3.us-west-1.amazonaws.com/A+Statistical+Framework+for+Analyzing+Specification+Resistance+to+Learnware-Inversion+Risks.pdf
>
> ---
>
> **Q2**: ... the analysis assumes a black-box adversary that can only interact with the model through queries.
>
> **A2**: Our choice of a black‑box adversary is motivated by two reasons:
>
> In the Learnware paradigm, reusing existing models often involves not a single model but also selecting **multiple models** to be composed. Consequently, the model $f^{\prime}$ that an adversary downloads may be formed by combining multiple base models $f_1, \cdots, f_n$ in a particular way. The diversity of composition mechanisms implies that the adversary may not have access to the exact structure of any specific base model $f_i$. Therefore, black-box attack analysis is essentially driven by this practical need.
>
> On the other hand, white-box attacks are dependent on the specific model $f$; for different model families the adversary's strategy may differ completely, which would confine the analysis to case-by-case studies. As the first work analyzing Learnware privacy, our primary goal is to establish a general theoretical analysis framework for Learnware privacy, and this framework can handle both **black-box and white-box attacks**. For black-box attacks in particular, we can provide a general theoretical formulation and derive general privacy guarantees for the existing RKME specification. This generality is more meaningful for Learnware privacy than model-specific case studies.
>
> ---
>
> **Q3**: ... could attacks such as model inversion attacks exploit both the model and the specification to recover information?
>
> **A3**: Thank you for this question. We would like to clarify that our framework is in fact designed to study exactly this type of threat, namely, attacks that jointly exploit both the model and the specification to recover information about the original training data. More specifically, we formulate this threat as a Learnware-Inversion attack.
>
> The notions of membership inference and attribute inference in our paper should be viewed as concrete privacy objectives instantiated within this general inversion framework, rather than as attack classes entirely separate from it. That is, they specify what kind of private information the adversary aims to recover, while the learnware-inversion/model-inversion viewpoint describes how the adversary exploits the released model and specification to do so.
>
> Therefore, our paper is not limited to analyzing a few isolated attack notions. Rather, it starts from the essential threat in the learnware setting, that an adversary only observes the model and the specification and attempts to infer information about the original training data, and then formalizes this threat through a unified framework, with membership inference and attribute inference serving as two representative and widely studied instantiations.
>
> ---
>
> If the reviewer finds that our responses have addressed the concerns, we would respectfully appreciate a reconsideration of the current score! We would also be very grateful for any further feedback or discussion!

---

> > ### Author Rebuttal · Reviewer_PhxA · 2026-04-03
> >
> > Thanks for the authors’ rebuttal and additional experiments, which have addressed my major concerns.

---

> > > ### Author Response · Authors · 2026-04-04
> > >
> > > Dear Reviewer PhxA,
> > >
> > > We sincerely thank the reviewer for the thoughtful follow-up and for taking the time to read our rebuttal and additional experiments so carefully! We are truly grateful that our clarification and supplementary results have addressed the reviewer’s major concerns. Your constructive feedback has been very valuable in helping us improve both the presentation and the technical discussion of the paper. We greatly appreciate your support!
> > >
> > > Best
> > >
> > > Authors

---

### Official Review · Reviewer_xNnm · 2026-03-19

**Soundness:** 2
**Presentation:** 2
**Significance:** 2
**Originality:** 2
**Overall Recommendation:** 4
**Confidence:** 3

**Summary:**

The paper investigates the privacy risks associated with releasing a ML model alongside a Reduced Kernel Mean Embedding (RKME) specification, an instance of the Learnware paradigm. In this context, the authors propose a "Learnware-Inversion" framework to quantify the privacy risk introduced by the specification. By analyzing the pre-image of the RKME mapping, they argue that the inherent data compression restricts an adversary's ability to infer training data, concluding that a specification of size $m = \sqrt{n}$ introduces small privacy risk.

**Compliance With Llm Reviewing Policy:**

Affirmed.

**Final Justification:**

Other than the empirical validation aspect, most of my concerns have been addressed.

**Key Questions For Authors:**

- The proofs for the key results assume that the model's prediction errors follow an exact Gaussian distribution. Modern machine learning models often exhibit skewed, heteroscedastic, or multimodal error distributions. How robust are your theoretical risk bounds if this Gaussian assumption is violated in practice?

**Limitations:**

No -  among the aspects authors could have mentioned are:

- lack of empirical validation of the bounds
- brittle distributional assumptions (i.e, Gaussian)

**Strengths And Weaknesses:**

**Strengths**

- The framing of the "Learnware-Inversion" game is interesting and seems like a contribution to the community, formalizing a threat model where the adversary has access to both the model and a distributional specification.

- The intuition that mapping a high-dimensional dataset to a lower-dimensional RKME somewhat acts as a privacy-preserving information bottleneck is interesting.

**Weaknesses**
While I really appreciate the theoretical setup of the work, I am having some trouble following a few of the steps in the appendices. I suspect there might be a few structural issues in the proofs that need clarification.

- I might be missing something, but it seems there is a significant leap in the geometric volume arguments used to bound the influence of the specification. The proof begins by citing Pan & Xu (2009), which establishes an isoperimetric inequality for a $C^2$ closed, strictly convex 2D plane curve. It then transitions to bounding the volume of an $(n-2m)$-dimensional manifold. I am not entirely sure how this 2D curve inequality generalizes to the high-dimensional pre-image manifold.

- I am also a bit uncertain about the claim that a fixed dataset $D$ yields a strictly unique RKME parameter set $Z$ and weights $\beta$.

- Given the non-convexity of the Gaussian kernel $k(x, z) = \exp(-\gamma\|x - z\|^2)$ with respect to $z$, I am hesitant to assume that the projection onto the $m$-dimensional subspace possesses a strictly unique global minimum. It would be very helpful if the authors could elaborate on how strict global convexity is guaranteed?

- Given that the theoretical bounds in the main writeup rely on complex, non-convex systems (and assume strict Gaussian error distributions), I belief that these bounds might behave differently in practice. The paper proposes several concrete attacks, but it provides no empirical evaluation. Including simulated Learnware-Inversion attacks on standard datasets to empirically verify that the risk remains below the 0.01 threshold would vastly strengthen the paper's claims.

---

> ### Author Rebuttal · Authors · 2026-03-30
>
> We sincerely thank the reviewer for the positive assessment of our theoretical setup, as well as for the careful reading of both the main paper and the appendix!
>
> Below, we provide detailed clarifications on the main concerns raised in the review, and hope that our responses will help address them.
>
> ---
>
> **Q1**: I might be missing something, ... ,  I am not sure how this 2D curve inequality generalizes to the high-dimensional pre-image manifold.
>
> **A1**: Thanks for your valuable feedback!
>
> Given an RKME, the set of original datasets that could generate it forms an $(n-2m)$-dimensional manifold. Among them, the datasets that may leak the target private information form an $(n-2m-1)$-dimensional submanifold, obtained by intersecting the original manifold with an $(n-2m-1)$-dimensional hyperplane. Therefore, estimating the privacy leakage risk reduces to comparing the volume of this submanifold with that of the original manifold.
>
> Our analysis is motivated by isoperimetric-type inequalities such as
>  $
>  P(E) \geq n \omega_n^{1/n} |E|^{\frac{n-1}{n}},
>  $
>  where $|E|$ is the $n$-dimensional volume and $P(E)$ is the $(n-1)$-dimensional boundary measure. The key idea is to impose suitable conditions under which a reverse form of such an inequality can be established, which then provides an upper bound on the measure of the lower-dimensional manifold. We do not directly follow the proof of Pan and Xu (2009). Rather, we adopt the type of structural condition they introduced and show that our RKME satisfies its natural higher-dimensional analogue (reverse Minkowski-type inequalities). This is why their result is relevant here: it provides a reasonable template for deriving the reverse isoperimetric guarantee needed in our analysis.
>
> ---
>
> **Q2 & Q3**: I am uncertain about that a fixed dataset $D$ yields a unique RKME, ..., how strict global convexity is guaranteed?
>
> **A2 & A3**: Thanks for your valuable feedback!
>
> We believe these two concerns stem from the same underlying question: although the Gaussian kernel leads to a non-convex optimization problem, why can we still ensure that a unique RKME corresponds to the original dataset?
>
> The key point is that (strong) convexity is only a sufficient condition for uniqueness of the minimizer, rather than a necessary one. Many non-convex functions still admit a unique global minimizer. For example, $ \cos x + \log(|x|+1) $ is non-convex and has countably infinitely many local minima, yet its global minimum is unique.
>
> For our problem, this conclusion is jointly supported by Lemma D.2 and Lemma D.5 in the appendix. In Lemma D.2, we show that whenever the RKME induces a nonzero compression, namely, the MMD distance between the RKME and the original sample KME is nonzero, uniqueness of the RKME follows from basic RKHS properties. Then, in Lemma D.5, we prove that the set of datasets for which no compression occurs is of measure zero in the original space. Combining these two results, we obtain that the RKME generated from the original dataset is unique almost everywhere.
>
> ---
>
> **Q4**: Including simulated Learnware-Inversion attacks on standard datasets to empirically would vastly strengthen the paper's claims.
>
> **A4**: We have added validation experiments. Due to the rebuttal word limit, detailed information about these supplementary experiments and results can be found in our response (**A1**) to Reviewer PhxA.
>
> ---
>
> **Q5**: How robust are your theoretical risk bounds if this Gaussian assumption is violated in practice?
>
> **A5**: We would like to clarify that the Gaussian assumption is used only in Theorem 3.4. Our other main results do not rely on any additional distributional assumption.
>
> For Theorem 3.4, the Gaussian assumption is mainly introduced to make the posterior-inference analysis tractable and to obtain an explicit closed-form characterization of the specification risk. This type of assumption is also common in privacy analysis when one aims to derive analytically interpretable bounds from error-based attacks. More importantly, handling fully general error distributions in posterior-based MIA remains challenging in all the theoretical analysis of such attacks. However, the general analysis pipeline remains applicable: one can replace the Gaussian likelihood with a more flexible estimate of the error distribution, such as a nonparametric density estimator or a richer parametric family (e.g., mixture models), and then perform the same posterior-based inference procedure.
>
> Therefore, we view Theorem 3.4 as a clean first-step theoretical result for a standard case, while the extension to more general error distributions is an important direction for future work. We will clarify this scope and limitation more explicitly in the revision.
>
> ---
>
> In the end, we take this opportunity to sincerely thank you for the careful review! If the reviewer feels that our clarifications have adequately addressed these concerns, we would be very grateful if the score could be reconsidered!

---

> > ### Author Rebuttal · Reviewer_xNnm · 2026-04-06
> >
> > My questions have mostly been addressed. I will increase my score.

---

> > > ### Author Response · Authors · 2026-04-06
> > >
> > > Dear Reviewer xNnm,
> > >
> > > We sincerely thank you for taking the time to read our rebuttal so carefully! We are very glad that our clarifications have addressed your concerns, and we are truly grateful for your decision to increase the score. This means a great deal to us and is a strong encouragement for our work!
> > >
> > > We will carefully revise the paper in future versions based on the points you raised, so as to further improve both the technical presentation and the overall quality of the paper.
> > >
> > > Finally, we would like to express our sincere appreciation again for your help and support!
> > >
> > > Best
> > >
> > > Authors

---

### Decision · Program_Chairs · 2026-04-30

**Decision:**

Accept (regular)

**Comment:**

This paper presents a novel and technically solid game-theoretic framework for quantifying the privacy risks associated with releasing machine learning models alongside Reduced Kernel Mean Embedding (RKME) specifications within the learnware paradigm. Reviewers appreciated the strong theoretical foundations but initially raised valid concerns regarding the lack of empirical validation, the reliance on Gaussian error distributions, and specific geometric and basis expansion assumptions in the proofs. During the rebuttal, the authors effectively addressed the majority of these critiques by providing supplementary experiments on real-world datasets, clarifying the measure-theoretic and mathematical steps in their appendices, and adequately explaining the scope of the black-box adversary setting. Although some reservations remain regarding the practicality of the Gaussian assumptions and the focus on distribution-level rather than worst-case privacy risks, the committee agrees that the analytical pipeline serves as a highly valuable first step for this problem space. Given the unified theoretical contribution and the authors' diligent improvements during the discussion period, the paper is recommended for acceptance.